# Development of cyclopeptide inhibitors of cGAS targeting protein-DNA interaction and phase separation

Xiaoquan Wang [1,7], Youqiao Wang [2,7], Anqi Cao[1], Qinhong Luo[1,3], Daoyuan Chen[4], Weiqi Zhao [5], Jun Xu [5], Qinkai Li[1,6], Xianzhang Bu [1,2] ✉ & Junmin Quan [1,6] ✉

Cyclic GMP-AMP synthase (cGAS) is an essential sensor of aberrant cytosolic DNA for initiating innate immunity upon invading pathogens and cellular stress, which is considered as a potential drug target for autoimmune and autoinflammatory diseases. Here, we report the discovery of a class of cyclopeptide inhibitors of cGAS identified by an in vitro screening assay from a focused library of cyclic peptides. These cyclopeptides specifically bind to the DNA binding site of cGAS and block the binding of dsDNA with cGAS, subsequently inhibit dsDNA-induced liquid phase condensation and activation of cGAS. The specificity and potency of one optimal lead XQ2B were characterized in cellular assays. Concordantly, XQ2B inhibited herpes simplex virus-1 (HSV-1)-induced antiviral immune responses and enhanced HSV-1 infection in vitro and in vivo. Furthermore, XQ2B significantly suppressed the elevated levels of type I interferon and proinflammatory cytokines in primary macrophages from *Trex1*[-/-] mice and systemic inflammation in *Trex1*[-/-] mice. XQ2B represents the specific cGAS inhibitor targeting protein-DNA interaction and phase separation and serves as a scaffold for the development of therapies in the treatment of cGAS-dependent inflammatory diseases.

The innate immune system recognizes foreign pathogens and cellular stress by sensing aberrantly modified or mislocalized nucleic acids[1,2]. The cGAS-STING pathway in the innate immune system is essential for cytosolic DNA sensing[3,4]. Cyclic GMP-AMP synthase (cGAS) is activated by sensing exogenous or endogenous double-stranded DNA (dsDNA) in the cytoplasm as a major mechanism by which cells detect cytosolic DNA[5]. Activated cGAS catalyzes the synthesis of 2′–5′, 3′–5′ cyclic GMP-AMP (2′3′-cGAMP) from GTP and ATP[6,7]. As a second messenger, 2′3′-cGAMP binds and activates the receptor stimulator of interferon genes (STING, also known as MITA, ERIS and MPYS)[8–11], which leads to recruitment and phosphorylation of downstream effectors TANK-binding kinase 1 (TBK1) and sequentially of interferon regulatory factor 3 (IRF3), resulting in the production of type I IFNs and proinflammatory cytokines[12–15].

cGAS plays a key role in antiviral response and antitumor immunity by inducing the expression of interferon-stimulated genes (ISGs)[16–19]. In contrast, abnormal activation of the cGAS-STING pathway has been shown to be a critical pathogenesis of autoimmune and autoinflammatory diseases[17,20–22], and even cancer[23,24]. Loss-of-function mutations in TREX1, the DNA exonuclease that is responsible for

[1]State Key Laboratory of Chemical Oncogenomics, Guangdong Key Laboratory of Chemical Genomics, Peking University Shenzhen Graduate School, Shenzhen 518055, China. [2]School of Pharmaceutical Sciences, SunYat-sen University, Guangzhou 510006, China. [3]Department of Pharmacy, Shenzhen Second People's Hospital (Shenzhen Institute of Translational Medicine), The First Affiliated Hospital of Shenzhen University, Shenzhen 518000, China. [4]School of Bioengineering, ZhuHai Campus of Zunyi Medical University, Zhuhai 519041, China. [5]Genetics and Metabolism Department, The Children's Hospital, School of Medicine, Zhejiang University, National Clinical Research Center for Child Health, Hangzhou 310052, China. [6]Shenzhen Bay Laboratory, Shenzhen 518055, China. [7]These authors contributed equally: Xiaoquan Wang, Youqiao Wang. ✉e-mail: phsbxzh@mail.sysu.edu.cn; quanjm@pku.edu.cn

degrading cytosolic DNA, lead to the accumulation of cytosolic dsDNA and aberrant activation of cGAS-STING pathway, subsequently trigger autoimmune diseases[25,26], such as Aicardi-Goutières syndrome (AGS)[27,28] or chilblain lupus[29] and systemic lupus erythematosus (SLE)[30,31]. Moreover, mitochondrial DNA stress invoked by a variety of cellular insults also promotes the inflammatory response through triggering cGAS activity[32–35]. The essential role of cGAS in innate immunity and autoinflammatory responses supports cGAS as a potential drug target for diseases associated with dysregulated cGAS activity.

Over recent years, considerable effort has been devoted to developing potent and specific cGAS inhibitors[36]. cGAS inhibitors based on diverse chemical scaffolds that target the active site of cGAS have been discovered by high-throughput screening assays and optimized by medicinal chemistry[37–39]. Moreover, a novel type of cGAS inhibitors targeting the dimeric interface of cGAS have been developed by a high-throughput virtual screening (HTVS) of ~1.7 million compounds and extensive medicinal chemistry optimization[40]. These inhibitors exhibit high in vitro biochemical activity but relatively low or moderate cellular activity, this discrepancy may be attributed to the potential competition of the high levels of intracellular ATP and GTP[39], or attributed to the relatively low cellular permeability of the inhibitors[38,40]. Disrupting dsDNA binding is another potential strategy to regulate the activity of cGAS[36]. Recently, a series of quinoline-containing antimalarial drugs and an approved drug suramin have been identified as cGAS inhibitors targeting the binding of cGAS to dsDNA[41,42]. However, both quinoline-containing antimalarial drugs and suramin have multiple potential targets[43,44], suggesting that they are non-specific cGAS inhibitors and may cause off-target effects in cells. Unlike conventional ligand-binding pockets of proteins, the solvent-exposed, highly positively charged, and flat features of protein–DNA interfaces make it challenging to develop small-molecule inhibitors with drug-like properties[45]. Macrocyclic peptides have proven to be a particularly useful tool to regulate protein-protein interactions (PPIs), another type of challenging interface for small-molecule inhibitors, for their high potency and selectivity[46], rendering macrocyclic peptides potential scaffolds for regulating protein-DNA interface.

In our ongoing efforts to develop macrocyclic peptide inhibitors targeting protein-protein interactions[47,48], we have also been trying to identify cGAS inhibitors targeting protein-DNA interface from our focused library of macrocyclic peptides. Here, we report the discovery of a class of macrocyclic peptide inhibitors of cGAS that target the protein-DNA interface between cGAS and dsDNA. These inhibitors bind with cGAS and disrupt the interaction between dsDNA and cGAS, further suppressing dsDNA-induced liquid phase condensation and activation of cGAS. We further demonstrate that one optimal lead XQ2B inhibited herpes simplex virus-1 (HSV-1)-induced antiviral immune responses and enhanced HSV-1 infection in vitro and in vivo. Furthermore, XQ2B significantly suppressed the elevated levels of type I interferon and proinflammatory cytokines in primary macrophages from *Trex1*[-/-] mice and systemic inflammation in *Trex1*[-/-] mice. XQ2B represents the specific cGAS inhibitor targeting protein-DNA interface, and serves as a scaffold for the further development of drugs for cGAS-related autoimmune and inflammatory diseases.

## Results

### Discovery of macrocyclic peptide inhibitors of cGAS
To screen direct cGAS inhibitors targeting protein-DNA interface from our developed focused library of macrocyclic peptides, we applied an RNA-based fluorescent biosensor for 2′,3′-cGAMP to detect cGAS activity in vitro (Supplementary Fig. 1a)[49]. The formation of 2′,3′-cGAMP from ATP and GTP was catalyzed by recombinant human cGAS enzyme (hcGAS-ΔN, residues 147–522, Supplementary Fig. 1b) in the presence of ISD (interferon stimulatory DNA, a 45-bp dsDNA), which was readily quantified by the RNA-based fluorescent biosensor and was

further confirmed by ion exchange chromatography (Supplementary Fig. 1c). The positive control quinacrine inhibited the activity of cGAS in a dose-dependent manner (Supplementary Fig. 1d), suggesting the validity of the established assay. Subsequently, a total of 90 macrocyclic peptides were screened at a concentration of 20 μM (Fig. 1a and Supplementary Fig. 1e), and 4 hits (XQ2, XQ8, XQ19, and XQ46) were obtained for further testing at a cut-off of 40% inhibition (Supplementary Table 1). To evaluate the inhibitory effect of the hits against cGAS in cells, we employed real-time quantitative PCR (qPCR) to examine mRNA levels of *IFNB1* in THP1 cells. The reported cGAS inhibitor RU.521 was used as the positive control, and the non-active peptide XQ28 in the screening assay was used as the negative control. Transfection of dsDNA markedly induced the expression level of *IFNB1*, while the dsDNA-induced mRNA levels of *IFNB1* was significantly inhibited by these hits (Fig. 1b). The inhibition of cGAS by these hits was further confirmed in the reporter cell line THP1-Lucia ISG (Invivogen) that expresses secreted luciferase induced by an IRF3-inducible ISG54 promoter (Fig. 1c). The discrepancy between the inhibitory effect of the hits in protein and cellular assays may be attributed to their different cellular permeability. XQ2 was selected for further evaluation because of the best potency in the cellular assays.

### XQ2 selectively inhibits dsDNA-induced signaling in human and murine cells
Luciferase reporter assay in the reporter cell line THP1-Lucia ISG showed that XQ2 inhibited dsDNA-induced signaling in THP1 cells in a dose-dependent manner with an inhibitory activity comparable to that of RU.521 (Fig. 2a). Consistently, the result of qPCR analysis revealed that XQ2 inhibited the expression level of *IFNB1* and *CXCL10*, as well as *IL6* induced by ISD or herpes simplex virus-1 (HSV-1) in a dose-dependent manner (Supplementary Fig. 2a, b). Notably, XQ2 selectively suppressed the mRNA level of *IFNB1* induced by the 45-bp dsDNA ISD or herpes simplex virus-1 (HSV-1), a classic DNA virus, but did not affect the expression of *IFNB1* induced by poly (I: C), a synthetic double-stranded RNA (dsRNA) analog that activates Toll-like receptor-3 (TLR3) and RIG-I/MDA-5 pathways (Fig. 2b). Furthermore, XQ2 also markedly suppressed the expression of *IFNB1* and *IL6* in primary human peripheral blood mononuclear cell (PBMCs) induced by ISD (Fig. 2c). We next evaluated the inhibitory effect of XQ2 against cGAS in murine fibroblast L929 cells, macrophage RAW264.7 cells and primary bone marrow-derived macrophages (BMDMs) from C57BL/6 mice. XQ2 at 10 μM significantly inhibited the expression of *Ifnb1* and its downstream gene *Cxcl10* in L929, RAW264.7, and BMDMs induced by ISD and HSV-1 (Supplementary Fig. 2c, d), and no cytotoxicity of XQ2 at the tested concentration was observed in both human and murine cell lines (Supplementary Fig. 2e–g), which suggested that XQ2 might serve as a selective inhibitor of both human and murine cGAS.

We next evaluated the effects of XQ2 on the downstream events of cGAS-STING pathway in THP1 cells stimulated by dsDNA. As expected, the immunoblotting analysis indicated that XQ2 markedly decreased phosphorylation of TBK-1 and IRF-3 induced by ISD (Fig. 2d). In addition, XQ2 also substantially suppressed the nuclear translocation of IRF-3 induced by ISD and HSV-1 (Fig. 2e). Consistently, the immunofluorescent analysis revealed that XQ2 significantly inhibited the nuclear translocation and phosphorylation of IRF-3 triggered by HSV-1 (Fig. 2f, g). Moreover, XQ2 significantly reduced ISD-induced autophagy as measured by the LC3-II formation (Supplementary Fig. 3a), and markedly suppressed ISD-induced cell death (Supplementary Fig. 3b). Altogether, the results demonstrated that XQ2 specifically blocked dsDNA-induced cGAS activation and cGAS-STING signaling.

### XQ2 directly binds with cGAS and blocks dsDNA binding
To examine the direct binding of XQ2 with cGAS, we expressed full-length human cGAS (hcGAS-FL) (Supplementary Fig. 4a) and

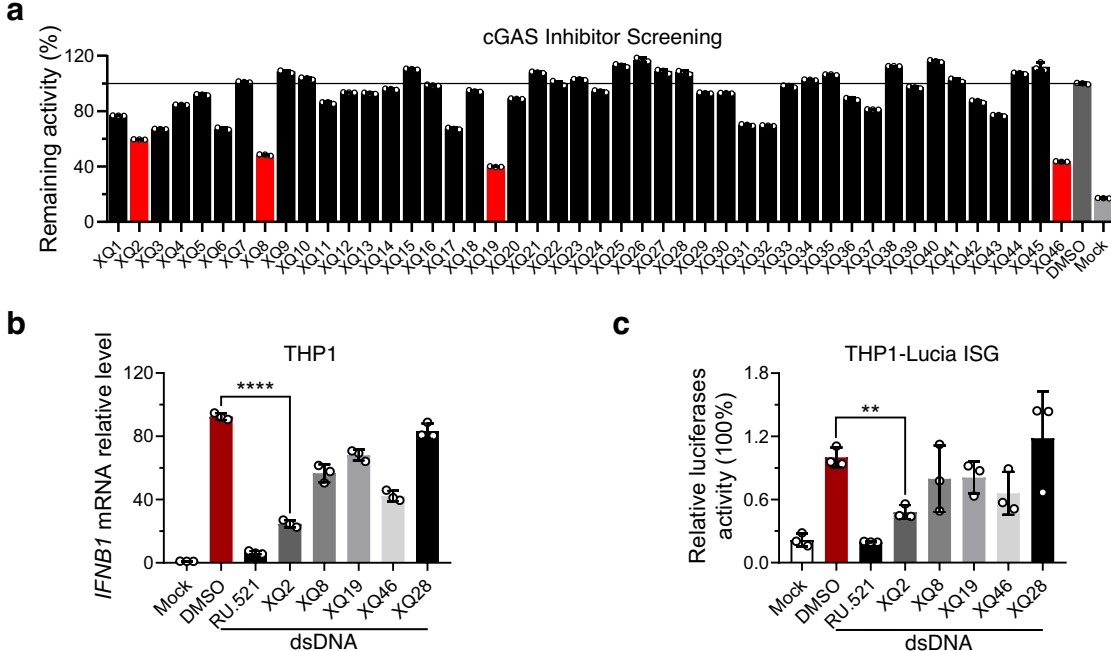

**Fig. 1 | Screening of cGAS inhibitors from macrocyclic peptide library. a** In vitro screening of cGAS inhibitors from macrocyclic peptide library by an RNA-based fluorescent biosensor. Hits were selected at a cutoff of 40% inhibition. **b** Potency of four hits (10 μM) was tested in dsDNA-stimulated THP1 cells in the presence of indicated molecule concentrations. RU.521 and XQ28 were used as the positive and negative controls, respectively. *IFNB1* mRNA in THP1 cells upon dsDNA stimulation for 6 h were measured by qPCR. $p < 0.0001$. **c** THP1 luciferase reporter cells were exposed to indicated compounds, and then stimulated with dsDNA for 24 h to promote type I interferon response. Type I interferon response was assessed by luciferase activity. $p = 0.0014$. Data are representative of three independent experiments in (**b, c**). (Data are presented as mean ± SD, $n = 3$ independent samples in (**a–c**), **$p < 0.01$, ****$p < 0.0001$ using one-way ANOVA with Dunnett's post hoc test). Source data are provided as a Source Data file.

conducted binding studies by surface plasmon resonance (SPR). SPR analysis indicated that XQ2 bound hcGAS-FL with an affinity ($K_d$) of $35.9 ± 10.0$ μM (Fig. 3a), and XQ2 also bound the truncated cGAS (hcGAS-ΔN, residues 147–522), without the disordered and positively charged N-terminus, with similar affinity to that of full-length cGAS (Supplementary Fig. 4b). Furthermore, the fluorescent polarization (FP) assay showed that XQ2 inhibited the binding of dsDNA with cGAS in a dose-dependent manner, with an IC50 value of $19.2 ± 6.7$ μM (Fig. 3b). To rule out the possibility that XQ2 interrupts the interaction between dsDNA and cGAS through intercalating dsDNA, we further examined the interaction between XQ2 and the 45-bp dsDNA ISD by microscale thermophoresis (MST). The result revealed no binding of XQ2 with ISD, suggesting no direct binding of XQ2 with DNA (Supplementary Fig. 4c). On the other hand, the relatively weak binding affinity of XQ2 with cGAS cannot fully explain the observed cellular efficacy of XQ2 against cGAS signaling (Fig. 2a, and Supplementary Fig. 2a, b). To address this discrepancy, we determined the binding affinity of XQ2 with cGAS by microscale thermophoresis (MST) in the presence or absence of ATP/GTP (Supplementary Fig. 4d). ATP/GTP significantly enhanced the binding affinity of XQ2 with cGAS ($Kd = 4.5 ± 1.6$ μM vs $Kd = 45.3 ± 26.9$ μM), suggesting that the high levels of intracellular ATP/GTP may facilitate the binding of XQ2 with cGAS. Consistently, ammonium sulfate (($NH_4)_2SO_4$), a mimics of phosphate groups that stabilizes the activation loop of cGAS[50], also markedly enhanced the binding affinity of XQ2 with cGAS (Supplementary Fig. 4e).

The aforementioned data suggest that the binding site of XQ2 may locate at the DNA binding site of cGAS. We tried to get the cocrystal structure of cGAS in complex with XQ2 to characterize the detailed binding mode of XQ2 on cGAS, but no complex structure was obtained. We thus carried out in silico docking to characterize the potential binding mode of XQ2 with cGAS. The docking structure revealed that XQ2 was packed to the

binding site of dsDNA on cGAS (Fig. 3c and Supplementary Fig. 5a). The favorable interaction between XQ2 and cGAS was characterized by the extensive hydrogen bonding and hydrophobic packing. The backbone carbonyl groups of XQ2 formed four hydrogen bonds with the sidechains of Asn210, Tyr214, His217 and Lys384 of cGAS. Notably, the positively charged sidechain of one arginine residue of XQ2 formed a strong salt-bridge with the negatively charged sidechain Asp191 of cGAS, which may contribute significantly to the interaction between XQ2 and cGAS. Consistently, replacement of the arginine residue with neutral alanine or negatively charged aspartate in the cyclic peptides markedly reduced the inhibitory activity against cGAS (Supplementary Fig. 5b and Supplementary Table 1). In addition, D191R mutation did not interfere with the binding of cGAS with dsDNA (Supplementary Fig. 5c), while markedly reduced the binding affinity of XQ2 with cGAS (Supplementary Fig. 5d) and the inhibitory effect of XQ2 against the binding of dsDNA with cGAS (Fig. 3d). It should be noted that the proposed binding mode of XQ2 on cGAS is speculative and needs further confirmed by the cocrystal structure of cGAS in complex with XQ2 in the future work.

## XQ2 inhibits DNA-induced liquid phase condensation of cGAS

DNA-induced liquid phase condensation of cGAS plays a crucial role in regulating the enzymatic activity and innate immune signaling[51]. Given the inhibitory effect of XQ2 against the binding of dsDNA with cGAS, we speculated that XQ2 may suppress DNA-induced liquid phase condensation of cGAS (Fig. 4a). Recombinant full-length cGAS and fluorescently labeled ISD (Cy3-ISD) readily formed liquid droplets when the concentration of each exceeded 100 nM (Supplementary Fig. 6a). Fluorescence recovery after photobleaching (FRAP) assay showed that the fluorescence of cGAS-Cy3-ISD condensates was efficiently recovered when bleaching was performed within 30 min after

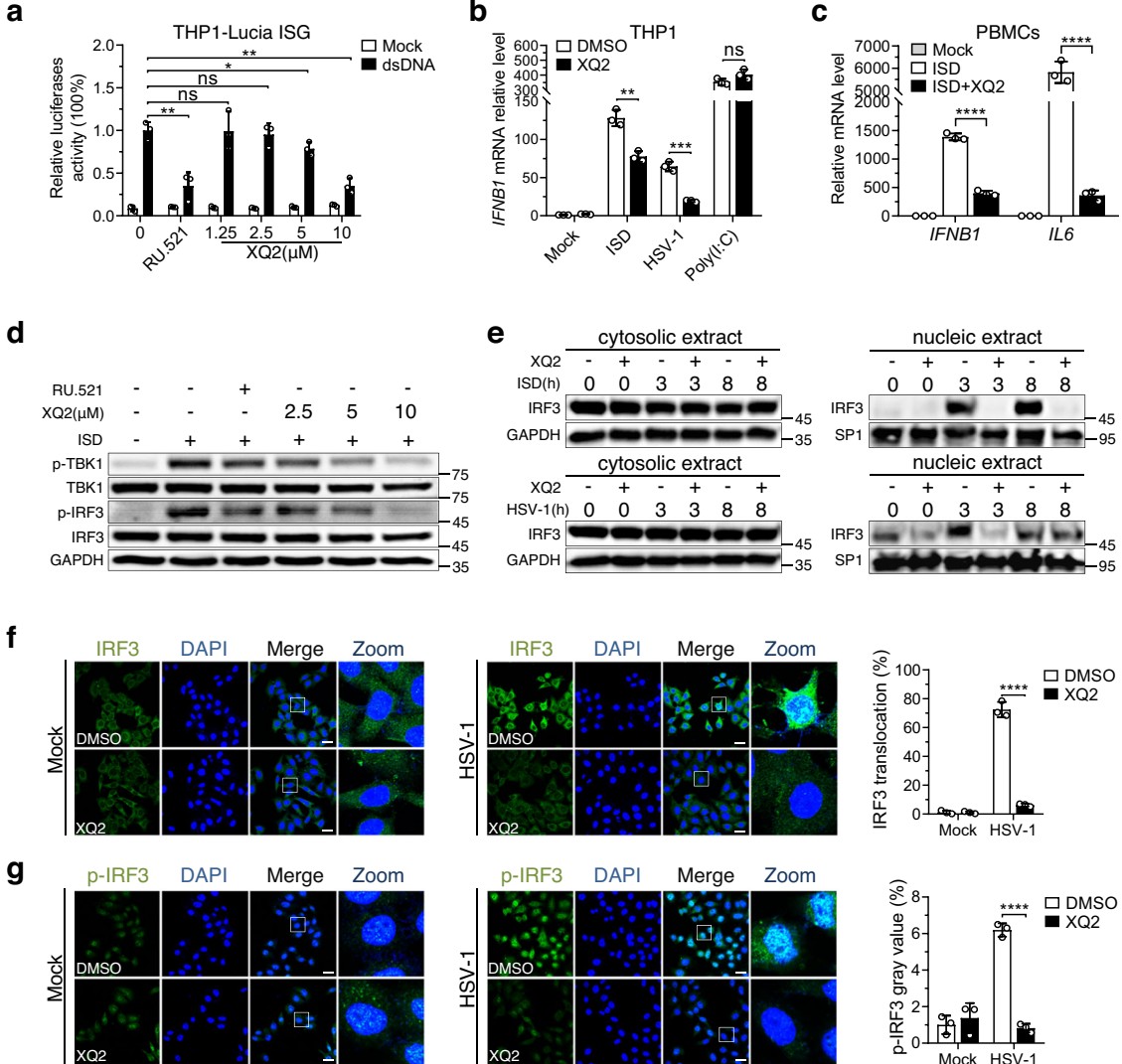

**Fig. 2 | XQ2 selectively inhibits dsDNA-induced signaling in human and murine cells. a** THP1 luciferase reporter cells were exposed to DMSO, RU.521 (10 μM), and the indicated doses of XQ2 for 3 h, and then stimulated with dsDNA for 24 h to promote type I interferon response. Type I interferon response was assessed by luciferase activity. $p = 0.0039$ (RU.521); $p = 0.0426$ (5 μM); $p = 0.0011$ (10 μM). **b** THP1 cells were pretreated for 3 h with DMSO or 10 μM XQ2, and then stimulated by transfection of ISD or HSV-1 infection or poly (I: C). Induction of *IFNB1* mRNA was measured by qPCR. $p = 0.0024$ (ISD); $p = 0.0003$ (HSV-1). **c** Primary human PBMCs were pretreated for 3 h with DMSO or XQ2 (10 μM), and then stimulated with ISD. Induction of *IFNB1* and *IL6* mRNA was measured by qPCR. $p < 0.0001$ (*IFNB1*); $p < 0.0001$ (*IL6*). **d** THP1 cells pretreated for 3 h with DMSO, RU.521 or the indicated doses of XQ2, followed by stimulation with dsDNA for 6 h, and then cell lysates were analyzed for phosphorylated IRF3 and TBK1 by immunoblotting. **e** THP1 cells were pretreated for 3 h with DMSO or XQ2 (10 μM), and then stimulated with ISD or

HSV-1 for indicated times. Cytoplasmic and nuclear fractions were extracted and immunoblotted with the indicated antibodies. SP1 and GAPDH were applied to indicate the accuracy of fractionation. **f, g** Hela cells were pretreated for 3 h with DMSO or XQ2 (10 μM), and then stimulated with HSV-1 for 6 h. Cells were immunostained with anti-IRF3 or anti-p-IRF3 antibody and imaged by confocal microscopy. Representative images (Left and Center); quantification of cells with nuclear IRF3 (Top and Right); quantification of cells with p-IRF3 (Bottom and Right). Scale bars represent 25 μm. $p < 0.0001$ (**f**); $p < 0.0001$ (**g**). Data are representative of three independent experiments with similar results in (**d**, **e**), or three independent experiments in (**a–c**, **f**, **g**). (Data are presented as mean ± SD, $n = 3$ independent samples in (**a–c**, **f**, **g**), ns, not significant, *$p < 0.05$, **$p < 0.01$, ***$p < 0.001$, ****$p < 0.0001$ using one-way ANOVA with Dunnett's post hoc test). Source data are provided as a Source Data file.

the initiation of phase separation (Fig. 4b, c). XQ2 efficiently suppressed DNA-induced liquid phase condensation of cGAS in a dose-dependent manner (Fig. 4d, e and Supplementary Fig. 6a–d), while RU.521, an active site inhibitor of cGAS, had no effect on DNA-induced liquid phase condensation of cGAS (Supplementary Fig. 6e, f). We further examined the effect of XQ2 on the formation of cGAS-DNA foci in cells. cGAS-deficient Hela cells were reconstituted with an enhanced green fluorescent protein (EGFP)-cGAS. EGFP-cGAS was diffused in the cytoplasm in the absence of ISD, and formed puncta in the cytoplasm with ISD, while XQ2 markedly inhibited the puncta formation of EGFP-cGAS with ISD (Fig. 4f). Collectively, XQ2 suppressed DNA-induced liquid phase condensation of cGAS in vitro and in cells.

## XQ2 and the optimized analogue XQ2B attenuate the host innate antiviral response

Given the essential role of cGAS in antiviral innate immunity, we next evaluated whether XQ2 antagonizes innate antiviral response by inhibiting cGAS activation. HSV-1 infection induced the production of IFN-β and CXCL10 in THP1 cells in a titer-dependent manner, which induced a robust immune response in higher titers. Pretreatment of XQ2 significantly inhibited the production of IFN-β and CXCL10 in THP1 cells upon HSV-1 infection (Fig. 5a, b), which was further confirmed in the reporter cell line THP1 Lucia ISG (Fig. 5c). Concordantly, XQ2 substantially facilitated HSV-1 infection of THP1 cells as reflected by the increased mRNA level of viral gene *HSV1(UL5)* (Fig. 5d).

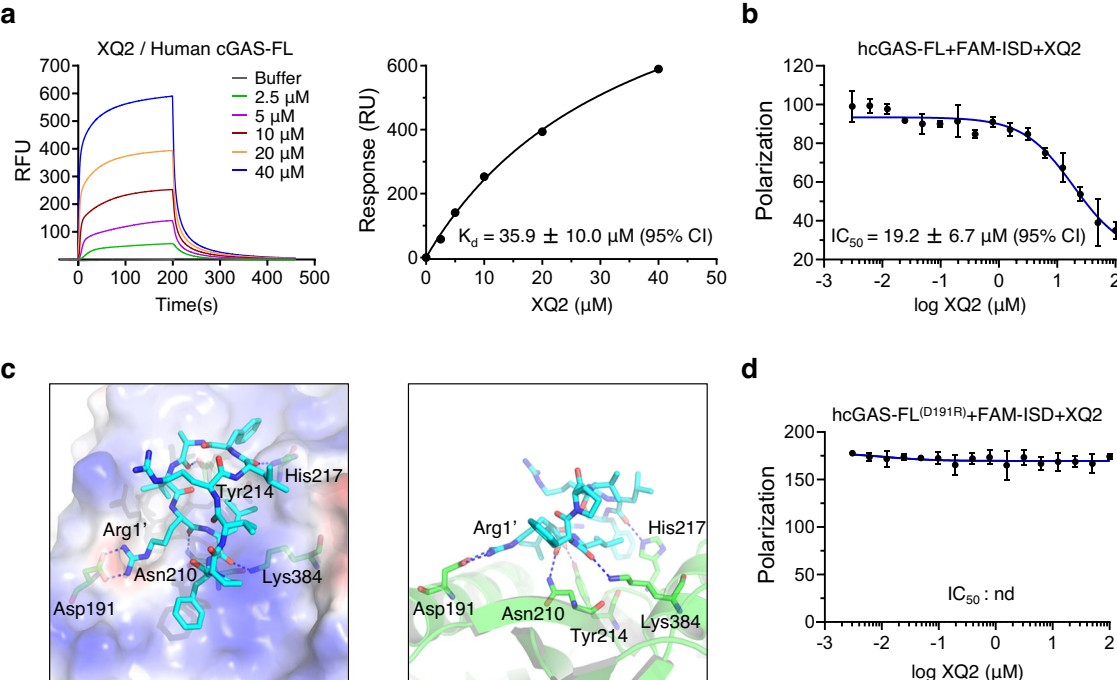

**Fig. 3 | XQ2 directly binds with cGAS. a** SPR analysis of the binding of XQ2 with full-length human cGAS (hcGAS-FL). The binding affinity ($K_d$) was determined by fitting the binding data to a simple one-to-one binding model. **b** Competition of FAM-ISD binding to hcGAS-FL with the indicated doses of XQ2, the polarization signal was detected by a microplate reader. **c** The docked structure of XQ2 with cGAS (PDB ID: 4O69), and the right panel indicated the detailed interactions between XQ2 and cGAS (right). XQ2 (cyan) and the contacted residues (green) of cGAS were shown as a stick model. Hydrogen bonds were indicated by dash blue lines. **d** Competition of FAM-ISD binding to hcGAS-FL (D191R) mutant with the indicated doses of XQ2, the polarization signal was detected by a microplate reader. Data are representative of three independent experiments in (**b**, **d**). (Data are presented as mean ± SD, $n = 3$ independent samples in (**b**, **d**). Source data are provided as a Source Data file.

Moreover, HSV-1 produced marked cytopathogenic effect (CPE) in THP1 cells in the presence of XQ2 (Fig. 5e), and the crystal violet staining based assay also demonstrated the promotion of XQ2 on HSV-1-induced cytopathogenic effect in mouse fibroblast L929 cells (Fig. 5f).

Despite the in vitro efficacy of XQ2 at the tested concentration, the in vivo application of XQ2 was limited by its cytotoxicity and low solubility at higher concentrations. The cytotoxicity of XQ2 is mainly attributed to its amphiphilic feature that may disrupt cell membrane integrity, and the hydrophobic patch of XQ2 results in low solubility[52–54]. To overcome this limitation, we modified XQ2 by replacing a valine residue with threonine to generate the analogue XQ2B (Fig. 6a and Supplementary Fig. 12), which would disrupt the hydrophobic patch and amphiphilic feature of XQ2. As expected, XQ2B exhibited less cytotoxicity and higher solubility compared to XQ2 (Supplementary Fig. 7a, b). Notably, XQ2B showed a comparable cGAS inhibitory effect as that of XQ2, while had no significant effect on signaling induced by STING agonists or TLR3 agonist such as poly (I: C) (Supplementary Fig. 7c–g). Consistently, XQ2B also markedly reduced ISD-induced autophagy and cell death (Supplementary Fig. 8a–d). Moreover, XQ2B also facilitated HSV-1 infection as XQ2 (Supplementary Fig. 8e). In consistent with the in vitro results, intravenous administration of XQ2B (10 mg/kg) significantly impaired the production of IFN-β and CXCL10 upon HSV-1 infection in mice (Fig. 6b, c), and thus promoted virus infection in the brain tissues of mice (Fig. 6d, e). Altogether, the in vitro and in vivo data demonstrate that XQ2B attenuates the host innate antiviral response.

**XQ2B suppresses systemic inflammation in *Trex1*⁻/⁻ mice.** Aberrant accumulation of cytosolic self-dsDNA caused by loss-of-function mutations in Trex1 leads to overactivation of cGAS-STING

pathway, which is associated with autoinflammatory diseases. To evaluate whether the optimized analogue XQ2B inhibits the overactivation of cGAS-STING pathway due to Trex1 deficiency, we first tested the effects of XQ2B on the expression of *Ifnb1*, *Cxcl10* and *Il6* in BMDMs harvested from *Trex1*⁻/⁻ mice (Supplementary Fig. 9a, b). XQ2B significantly suppressed the expression of *Ifnb1*, *Cxcl10* and *Il6* in *Trex1*⁻/⁻ BMDMs (Fig. 7a), and markedly inhibited the expression of these genes in mouse L929 and human THP1 cells with *Trex1*-siRNA (Supplementary Fig. 9c–f). We next evaluated the anti-inflammatory effect of XQ2B in *Trex1*⁻/⁻ mice. XQ2B (10 mg/kg) was intravenously injected into mice every other day. Six animals from each group were sacrificed on day seven to assess tissue pathology, and the rest (10 mice per group) were used to analyze survival for up to 11 days (Fig. 7b). As shown in Fig. 7c, 3 of 10 untreated *Trex1*⁻/⁻ mice died, while none of the 10 mice treated with XQ2B died during treatment ($p = 0.035$). Mice were killed and tissues were collected on day seven for histology and mRNA analysis. Hematoxylin and eosin (H&E) analysis indicated that *Trex1*⁻/⁻ mice developed profound inflammation in heart, stomach, tongue, kidney, and skeletal muscle compared to wildtype mice, while XQ2B significantly attenuated the inflammation (Fig. 7d). Consistently, the qPCR analysis of heart tissue also demonstrated that *Trex1* deficiency increased the expression of *Ifnb1*, *Cxcl10* and *Il6*, and XQ2B treatment significantly suppressed the mRNA level of *Ifnb1*, *Cxcl10* and *Il6* caused by *Trex1* deficiency (Fig. 7e). Moreover, the serum level of antinuclear antibody was increased in *Trex1*⁻/⁻ mice, which was markedly reduced by XQ2B treatment (Fig. 7f). Altogether, these results demonstrated that XQ2B efficiently attenuated systemic inflammation in *Trex1*⁻/⁻ mice, highlighting the therapeutic potential of cGAS-specific inhibitors targeting protein-DNA interface in cGAS-related autoimmune diseases.

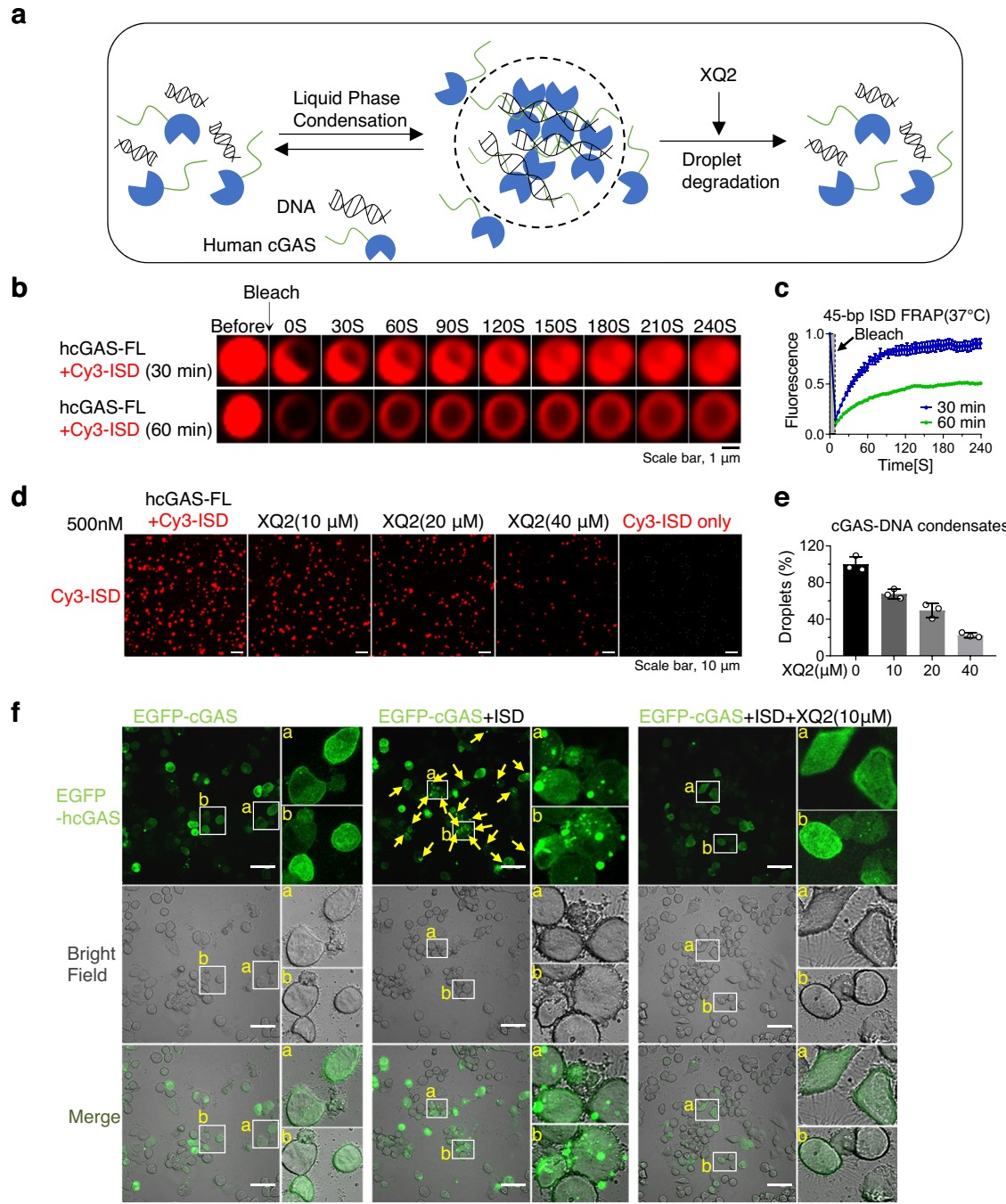

**Fig. 4 | XQ2 inhibits DNA-induced liquid phase condensation of cGAS.**
**a** Schematic of the inhibition of XQ2 against the liquid phase condensation of cGAS-DNA. **b**, **c** FRAP of cGAS-DNA condensates. Bleaching was performed at the indicated time points (30 and 60 min) after mixing of hcGAS-FL (1 μM) and Cy3-ISD (1 μM), and the recovery occurred at 37 °C. Time 0 s indicates the start of recovery after photobleaching. FRAP of cGAS-DNA droplets was quantified by fluorescence intensity. **d**, **e** The liquid phase condensation of hcGAS-FL (500 nM) protein and Cy3-ISD (500 nM) in the presence of either DMSO or the indicated doses of XQ2 for 30 min at 37 °C. The images shown in (**d**) are representative of all fields in the well.

Quantification of condensation was shown by the relative number of droplets. **f** cGAS-knockout Hela cells reconstituted with EGFP-hcGAS-FL were stimulated by transfection of ISD in the presence of either DMSO or XQ2 (10 μM) for 4 h. Representative imaging analysis by confocal microscope. Individual cells and cGAS–DNA puncta (Boxes) are enlarged. These images represent at least five fields examined. Data are representative of three independent experiments with similar results in (**f**), or three independent experiments in (**b**–**e**). (Data are presented as mean ± SD, *n* = 3 independent samples in (**c**, **e**). Source data are provided as a Source Data file.

## Discussion

Aberrant activation of cGAS-STING signaling has been shown to be closely associated with autoimmune and inflammatory diseases such as rheumatoid arthritis (RA)[55], Aicardi Goutières syndrome (AGS)[27,28], systemic lupus erythematosus (SLE)[30,31] and COVID-19[56]. In addition, emerging evidence suggests the pro-tumor roles of the cGAS-STING

pathway in tumor initiation, development, and metastasis[57]. The cGAS-STING pathway is therefore considered as a potential therapeutic target in inflammatory diseases. However, the development of cGAS-STING antagonists in the clinic remains challenging due to the difficulty of achieving high potency and selectivity[36]. In the present study, we identified a class of macrocyclic peptide inhibitors of cGAS

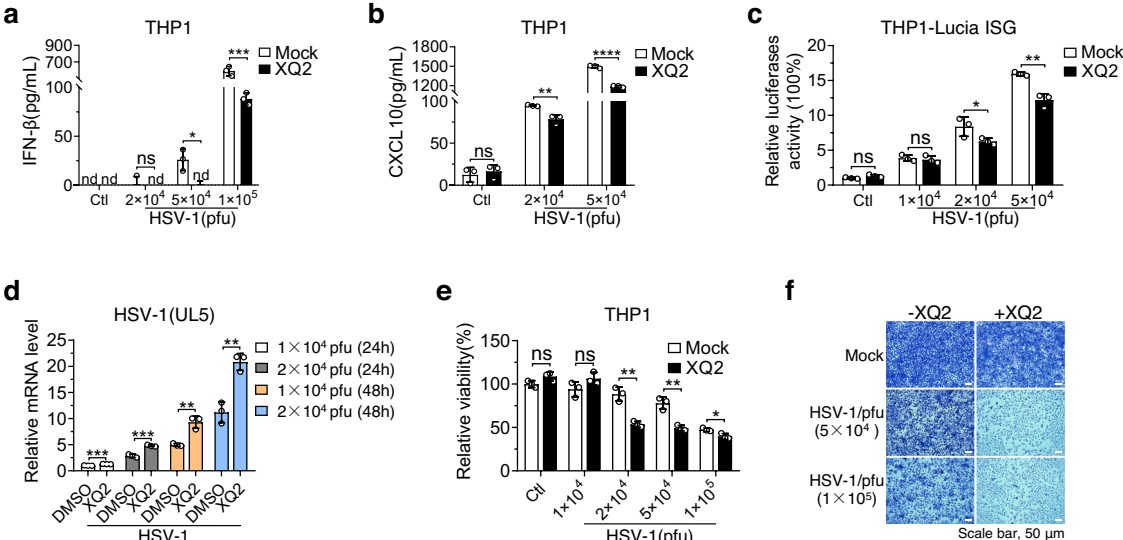

**Fig. 5 | XQ2 attenuates the host antiviral defenses. a**, **b** THP1 cells were pre-treated for 3 h with DMSO or XQ2 (10 µM), and then stimulated by HSV-1 infection for 24 h. Induction of IFN-β (**a**) and CXCL10 (**b**) proteins was measured by ELISA. **a** $p = 0.0264$ ($5 \times 10^4$ pfu); $p = 0.0001$ ($1 \times 10^5$ pfu), **b** $p = 0.0064$ ($2 \times 10^4$ pfu); $p < 0.0001$ ($5 \times 10^4$ pfu). **c** THP1 luciferase reporter cells were exposed to DMSO or XQ2 (10 µM), and then stimulated by the indicated titer of HSV-1 infection for 24 h. Type I interferon response was indicated by luciferase activity. $p = 0.043$ ($2 \times 10^4$ pfu); $p = 0.0018$ ($5 \times 10^4$ pfu). **d** THP1 cells were pretreated for 3 h with DMSO or XQ2 (10 µM), and then stimulated by HSV-1 infection for indicated titer and times. Induction of *HSV-1(UL5)* mRNA was measured by qPCR. $p = 0.0008$ ($1 \times 10^4$ pfu, 24 h); $p = 0.0007$ ($2 \times 10^4$ pfu, 24 h); $p = 0.0035$ ($1 \times 10^4$ pfu, 48 h); $p = 0.0029$ ($2 \times 10^4$ pfu, 48 h). **e** THP1 cells were pretreated for 3 h with DMSO or XQ2 (10 µM), followed by the indicated titer of HSV-1 infection. The cell viability was measured by CCK-8 assay. $p = 0.0024$ ($2 \times 10^4$ pfu); $p = 0.003$ ($5 \times 10^4$ pfu); $p = 0.0247$ ($1 \times 10^5$ pfu). **f** L929 cells pretreated for 4 h with DMSO or XQ2 (10 µM), followed by HSV-1 infection. The proliferation of cells was examined by crystal violet staining. Data are representative of three independent experiments with similar results in (**f**), or three independent experiments in (**a**–**e**). (Data are presented as mean ± SD, $n = 3$ independent samples in (**a**–**e**), ns, not significant, *$p < 0.05$, **$p < 0.01$, ***$p < 0.001$, ****$p < 0.0001$ using one-way ANOVA with Dunnett's post hoc test). Source data are provided as a Source Data file.

targeting the interface between cGAS and dsDNA. The SPR and FP experiments demonstrated that XQ2, one of these inhibitors, directly bound with cGAS and disrupted the interaction between dsDNA and cGAS, and subsequently inhibited dsDNA-induced activation of the cGAS-STING pathway. In previous studies[41,58], quinoline-containing antimalarial drugs, such as quinacrine, chloroquine and hydroxy-chloroquine, have been shown to disrupt the binding of dsDNA with cGAS and inhibit the enzymatic activity of cGAS. Moreover, the poly-anionic drug suramin also functioned as a cGAS inhibitor through displacing the bound DNA from cGAS[42]. On the other hand, currently, no experimental data supported the direct interaction of quinoline-containing antimalarial drugs or suramin with cGAS. In addition, qui-nacrine is a known DNA intercalator[59], which suggests that quinoline-containing antimalarial drugs may bind with dsDNA rather than bind with cGAS to disrupt the interaction between cGAS and dsDNA. Fur-thermore, suramin is also an inhibitor of multiple targets including G protein-coupled receptors, protein tyrosine phosphatases, and topoi-somerases as well as a large number of enzymes involved in DNA and RNA synthesis and modification, highlighting potential off-target effects beyond cGAS inhibition[44]. In this regard, XQ2 and its macro-cyclic peptide analogues may represent the first type of specific cGAS inhibitors targeting protein-DNA interface.

Combined in silico docking and mutagenic analysis revealed that XQ2 may bind to the DNA-binding site of cGAS. Several residues including Asn210, Tyr214, and Lys384 that form hydrogen bonds with the backbone of XQ2 are also involved in the binding with dsDNA[50,60,61]. The salt bridge formed between Asp191 of cGAS and the arginine side chain of XQ2 is essential for the binding of XQ2 with cGAS, though Asp191 is not involved in the direct binding of dsDNA with cGAS. Consistently, mutation of Asp191 to arginine in cGAS abrogated the binding of cGAS with XQ2, and replacement of the arginine residue with alanine or aspartate in the cyclic peptides markedly reduced the inhibitory activity against cGAS. Moreover, the critical residues of

cGAS involved in the interactions with XQ2 are conserved between human and murine (Supplementary Fig. 10), which might explain the similar inhibitory effect of XQ2 against cGAS in human and murine cell lines. Notably, the docking structure revealed that three of the four hits, including XQ2, XQ8, and XQ19 that contain a Val-Arg1'-Leu motif, share similar hydrophobic and electrostatic interactions with cGAS. On the other hand, another hit XQ46 with a substitution of Arg1' by a large hydrophobic Leu residue exhibited a comparable inhibitory activity as that of XQ2, XQ8 and XQ19, in which the impaired salt bridge between XQ46 and cGAS may be compensated by the hydrophobic interactions between Leu1' of XQ46 and Lys187 and Leu208 in cGAS (Supplementary Fig. 5b). This result underscores the challenge of achieving a consensus model to understand the structure activity relationship (SAR) for all the peptides in the focused library. Therefore, it should be noted that cocrystal structure of XQ2 with cGAS is necessary to understand the detailed binding mode of XQ2 on cGAS.

Recently, liquid-liquid phase separation (LLPS) has been recog-nized as an important framework for spatiotemporally regulating cel-lular cGAS responses[62]. Besides the DNA binding site of the NTase domain, the positively charged intrinsically disordered N terminus of cGAS facilitates multivalent interactions between cGAS and dsDNA, which results in liquid phase condensation of the complex of cGAS and dsDNA through liquid-liquid phase separation[51,63]. cGAS-DNA con-densates not only enhance the enzymatic activity of cGAS, but also inhibit dsDNA degradation mediated by three-prime repair exonu-clease 1 (TREX1)[64]. XQ2 blocked dsDNA-induced liquid phase con-densation of cGAS through disrupting the interaction between dsDNA and cGAS, which suppressed the enzymatic activity of cGAS, and facilitated TREX1-mediated degradation of cytosolic dsDNA. This mode of action may partially account for the stronger cellular inhibi-tory effect of XQ2 against cGAS activation compared to its binding affinity with cGAS and inhibitory activity against the binding of cGAS with dsDNA in biochemical assays. In addition, enhanced binding of

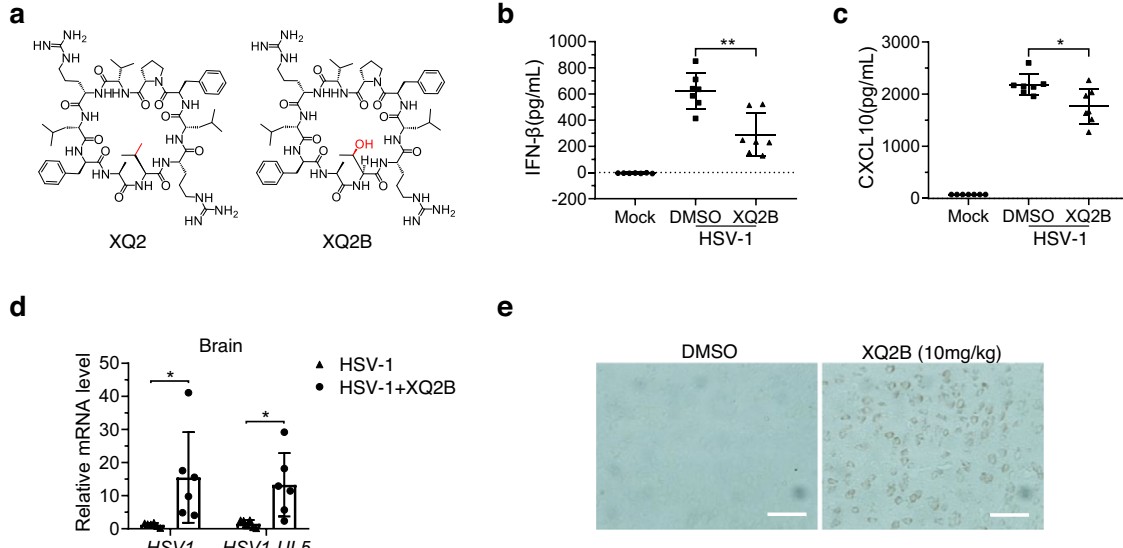

**Fig. 6 | XQ2B attenuates the host innate antiviral response in vivo. a** Chemical structure of XQ2 and XQ2B. **b**, **c** WT mice ($n = 7$) were injected intravenously with DMSO or XQ2B (10 mg/kg) for 3 h, and then administrated intravenously with HSV-1 at $1 \times 10^7$ pfu per mouse for 6 h. Serum from mice was collected for ELISA analysis of the levels of IFN-β (**b**) and CXCL10 (**c**). $p = 0.0013$ (**b**); $p = 0.0175$ (**c**). **d** Brains from mice ($n = 6$) in (**b**, **c**) were collected for qPCR analysis of the induction of *HSV-1* and *HSV-1(UL5)* mRNA. $p = 0.0287$ (*HSV-1*); $p = 0.0136$ (*HSV1-UL5*). **e** Brains from mice in

(**b**, **c**) were collected for immunohistochemistry (IHC) analysis with an anti-HSV-1 antibody. Tissue sections were visualized using microscopy. Scale bar, 25 µm. Data are representative of two independent experiments in (**b**–**e**). (Data are presented as mean ± SD, $n = 7$ mice per condition in (**b**, **c**); $n = 6$ mice per condition in (**d**), *$p < 0.05$, **$p < 0.01$ using one-way ANOVA with Dunnett's post hoc test). Source data are provided as a Source Data file.

XQ2 with cGAS by the high levels of intracellular ATP/GTP may also contribute to the higher cellular activity of XQ2. In contrast, active site inhibitors of cGAS with diverse chemical scaffolds, such as RU.521, G150 and PF-06928215, generally exhibited relatively weaker cellular activities compared to in vitro activities in biochemical assays[37–39]. The authors speculated that inhibiting the active site might require more potent compounds due to the competition of high intracellular levels of ATP and GTP[39], and due to the necessity to overcome the cellular permeation and transporter efflux[38]. Furthermore, cGAS-DNA liquid condensates may represent another barrier to modulating the cellular efficacy of cGAS inhibitors, since liquid-liquid phase separation has been emerging as a novel mechanism of understanding the pharmacodynamics for cancer therapeutics[65–67].

Both XQ2 and XQ2B were derived from Gramicidin S (GS)[47,48], one of the oldest commercially used cyclic peptide antibiotics that has robust antibacterial activity against both Gram-positive and Gram-negative bacterial strains. The biological activity and potential toxicity of GS are associated with its amphiphilic feature[68]. Replacement of a valine residue in XQ2 with threonine generated XQ2B, which disrupted the amphiphilic feature of XQ2 and resulted in less cytotoxicity and higher solubility, while kept the cellular inhibitory effect of cGAS and efficiently suppressed systemic inflammation in *Trex1-/-* mice. On the other hand, given diverse biological activities of derivatives of GS[68], the selectivity of XQ2B should be assessed more comprehensively in different systems. In the present study, we showed that XQ2B selectively inhibited dsDNA-induced signaling, and did not significantly affect the signaling induced by STING agonists or TLR3 agonists, but the effects of XQ2B on other immunogenic stimuli should be evaluated in future studies. Moreover, though XQ2B had improved safety and solubility properties compared to XQ2, it should be noted that a single dose of XQ2B used in the animal experiments is limited to fully characterize the safety and pharmacokinetic profiles of XQ2B. Despite the exhibited efficacy of XQ2B in the *Trex1-/-* mouse study, the preliminary pharmacokinetic study suggested that the dosing of XQ2B (10 mg/kg, q.o.d) may be suboptimal (Supplementary Fig. 11). Further evaluation of different doses and routes of administration of XQ2B in animals will

likely facilitate the future optimization of XQ2B to improve the translational potential of GS derivatives in cGAS-dependent diseases.

In summary, we identified a class of macrocyclic peptide inhibitors of cGAS targeting protein-DNA interaction and liquid-liquid phase separation. These inhibitors bound with cGAS and disrupted the interaction between dsDNA and cGAS, thus suppressed dsDNA-induced liquid phase condensation and cGAS activation. One optimal lead XQ2B inhibited herpes simplex virus 1 (HSV-1)-induced antiviral immune responses and enhanced HSV-1 infection in vitro and in vivo. Furthermore, XQ2B significantly suppressed the elevated levels of type I interferon and proinflammatory cytokines in primary macrophages from *Trex1-/-* mice and systemic inflammation in *Trex1-/-* mice. XQ2B represents the specific cGAS inhibitor targeting protein-DNA interface, and serves as a scaffold for the further development of drugs for cGAS-related autoimmune and inflammatory diseases.

## Methods
### Macrocyclic peptide library
The macrocyclic peptide library was derived from Gramicidin S[47,48]. Dissolving each compound with DMSO to the final concentration of 100 mM, and then diluted with reaction buffer (40 mM Tris-HCl, pH 7.5, 100 mM NaCl, 10 mM MgCl$_2$) until the final concentration is 100 µM, which is used as the stock for inhibitor screening and stored at −80 °C.

### Recombinant cGAS protein expression and purification
The gene encoding full-length human cGAS (hcGAS-FL) and N-terminally truncated human cGAS (hcGAS-ΔN, residues 147–522) were inserted into the pET-28a vector carrying an N-terminal 6xHis-SUMO tag. hcGAS-FL (D191R) mutant was generated by Fast Mutagenesis System (TransGen Biotech) using the pET-28a-hcGAS-FL as the template and confirmed by sequencing. The primer sequences used for cloning recombinant cGAS can be found in Supplementary Table S2. All proteins were overexpressed and purified from *Escherichia coli* strain BL21 (DE3). Bacteria were harvested and then lysed by sonication in lysis buffer (25 mM HEPES, pH 7.5, 300 mM

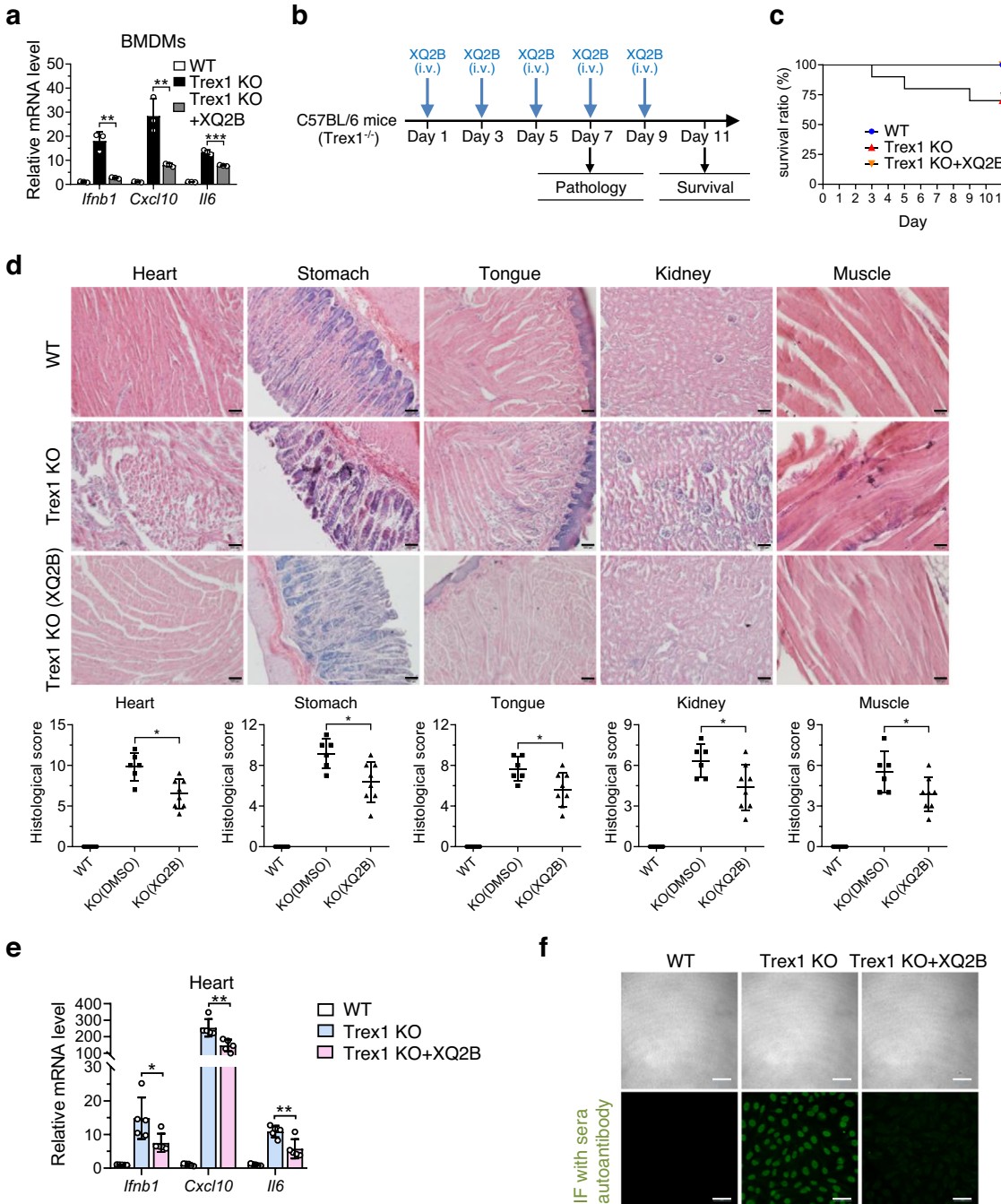

**Fig. 7 | XQ2B suppresses systemic inflammation in *Trex1*−/− mice. a** *Trex1*−/− BMDMs were treated with DMSO or XQ2B (10 μM) for 24 h, and induction of *Ifnb1*, *Cxcl10* and *Il6* mRNA was measured by qPCR. Fold changes are relative to WT BMDM group. $p = 0.002$ (*Ifnb1*); $p = 0.008$ (*Cxcl10*); $p = 0.0005$ (*Il6*). **b** Schematic representation illustrating the experimental design in the *Trex1*−/− mice model of autoimmune diseases. **c** Survival curves of WT and *Trex1*−/− mice treated with DMSO or XQ2B (10 mg/kg) every other day for 11 consecutive days ($n = 10$ mice per group). $p = 0.0352$. **d** WT mice ($n = 6$) or *Trex1*−/− mice ($n = 6$) were injected intravenously with DMSO or 10 mg/kg XQ2B every other day for 7 consecutive days. Representative H&E-stained tissue sections from WT or *Trex1*−/− mice treated with DMSO or XQ2B. The panels are at ×20 magnification. Scale bar, 100 μm. $p = 0.0139$ (Heart);

$p = 0.0139$ (Stomach); $p = 0.0274$ (Tongue); $p = 0.0315$ (Kidney); $p = 0.0139$ (Muscle). **e** heart tissues from WT, *Trex1*−/− and *Trex1*−/− (XQ2B) mice ($n = 5$) were collected and induction of *Ifnb1*, *Cxcl10* and *Il6* mRNA was measured by qPCR. $p = 0.0415$ (*Ifnb1*); $p = 0.0054$ (*Cxcl10*); $p = 0.0087$ (*Il6*). **f** Antinuclear antibodies in WT, *Trex1*−/− and *Trex1*−/− (XQ2B) serum were detected using antinuclear antibody antigen substrate slide kit. Scale bar, 50 μm. Data are representative of two independent experiments with similar results in (**f**), or three independent experiments in (**a**) or two independent experiments in (**d**, **e**). (Data are presented as mean ± SD, $n = 3$ independent samples in (**a**); $n = 6$ mice per condition in (**d**); $n = 5$ mice per condition in (**e**), $*p < 0.05$, $**p < 0.01$, $***p < 0.001$ using one-way ANOVA with Dunnett's post hoc test). Source data are provided as a Source Data file.

NaCl, 10% (v/v) glycerol, 5 mM MgCl₂, 1 mM PMSF, 15 mM imidazole, 20 μg/mL DNase I, 1 mM TCEP). The fusion proteins were purified through Ni²⁺-NTA resin beads (Qiagen). The 6xHis-SUMO tag was removed by SUMO protease (ULP1) at 4 °C overnight. Cleaved

proteins were further purified over a 5-ml HiTrap Heparin column (GE Healthcare), and the fractions were collected and concentrated. The final sample of hcGAS-FL, hcGAS-ΔN and hcGAS-FL (D191R) contains about 8 mg ml⁻¹ protein.

## Preparation of dsDNA for cGAS activity assays

DNA oligonucleotides and fluorescently labeled DNA oligonucleotides were synthesized by Sangon Biotech. Fully complementary strands were annealed in annealing buffer (10 mM Tris-HCl, pH 7.5, 50 mM NaCl, 1 mM EDTA) by first incubating at 95 °C for 2 min followed by ramping down from 95 °C to 25 °C at 0.05 °C s⁻¹. Complete annealing was verified by agarose gel electrophoresis. The sequences used for annealing are listed in Supplementary Table S2.

## Biosensor-based high-throughput analysis of cGAS activity in vitro

The high-throughput screening (HTS) experiments of cGAS activity are based on the method previously described[49]. To initiate the enzyme reaction, ISD was added to the reaction solution (40 mM Tris-HCl, pH 7.5, 100 mM NaCl, 10 mM MgCl2) containing hcGAS-ΔN, ATP (Sangon Biotech, Cat# A600311) and GTP (Sangon Biotech, Cat# A620332) in a total volume of 20 ul with the final concentration of 0.5 μM, 1.5 μM, 1 mM, 1 mM, respectively, and incubated for 2 h at 37 °C. For HTS experiments, compounds were added to the same reaction conditions as described above with the final concentration of 20 μM. Z′ factors were calculated from three replicates in the 384-well plate[69].

## Cell culture

THP1, Jurkat, Hela, L929 and RAW264.7 cells were purchased from the Shanghai Cell Bank of the Chinese Academy of Sciences (Shanghai, China). THP1-Lucia ISG cells were purchased from Invivogen. cGAS⁻/⁻ Hela cells were a kind gift from Dr. Zhengfan Jiang (School of Life Science, Peking University, Beijing, China). Human peripheral blood mononuclear cells (PBMCs) were purchased from LDEBIO (Guangzhou, China). THP1, THP1-Lucia ISG cells, Jurkat and PBMCs were cultured in RPMI 1640 (Gibco) supplemented with 10% heat-inactivated fetal bovine serum (FBS) (Gbico) and 0.1% Normocin (InvivoGen, Cat# ant-nr) and 0.1% mycoplasma elimination reagent (Yeasen, Cat# 40607ES03) at 37 °C with 5% CO₂. Hela, cGAS⁻/⁻ Hela, L929 and RAW264.7 cells were cultured in Dulbecco's modified Eagle's medium (DMEM) (Gbico) supplemented with 10% FBS and 0.1% Normocin (InvivoGen, Cat# ant-nr) and 0.1% mycoplasma elimination reagent (Yeasen, Cat# 40607ES03) at 37 °C with 5% CO₂.

## Antibodies

Rabbit anti-TBK1 (Cat# 38066, 1:1000), rabbit anti-phospho-TBK1 (Cat# 5483, 1:1000), rabbit anti-IRF3 (Cat# 11904, 1:1000), rabbit anti-phospho-IRF3 (Cat# 37829, 1:1000), rabbit anti-LC3A/B (Cat# 12741, 1:1000) and rabbit anti-GAPDH (Cat# 5174, 1:1000) were from Cell Signaling Technology. Rabbit anti-HSV-1 (Cat# ab9533, 1:100) was from Abcam. Rabbit anti-SP1 (Cat# 21962-1-AP, 1:1000) and mouse anti-alpha tubulin (Cat# 66031-1-Ig, 1:1000) were from Proteintech. Mouse anti-TREX1 (Cat# sc-133112, 1:1000) was from Santa Cruz Biotechnology.

## Real-time quantitative PCR

The indicated cells were stimulated with either ISD (Sangon Biotech), HSV-1 (Provided by Dr. Zhengfan Jiang), HSV60mer (Sangon Biotech), SATE-cddA[70], 2'3'-cGAMP (Invivogen, Cat# tlrl-nacga23-1), ADU-S100 (MedChemExpress, Cat# HY-12885A) or Poly(I:C) (Invivogen, Cat# tlrl-pic) after XQ2 or XQ2B treatment. Total RNA was isolated from indicated cells that incubated with the drugs for indicated time or mice tissues by using NucleoZOL reagent (MACHEREY-NAGEL) according to the manufacturer's instructions. The quantifications of mRNA levels were carried out by real-time PCR using a PerfectStart® Uni RT&qPCR Kit (TransGen Biotech) on the QuantStudio 5 qPCR machine (Applied Biosystems). Samples were carried out in triplicate and the target $C_T$ values were normalized to *GAPDH* $C_T$ values. Primers of qPCR used in this study can be found in Supplementary Table S2.

## Luciferase reporter assay for THP1

THP1-Lucia ISG cells (Invivogen) were seeded into a 96-well plate and incubated with RU.521 (Invivogen) or XQ2 for the indicated time, and then transfected with dsDNA by lipofectamine 2000 (Invitrogen) or infected by HSV-1 with the indicated titer for 24 h. The supernatant was collected and 50 μL QUANTI-Luc Luciferase reagent (Invivogen) was added to 20 μL supernatants per well, and luciferase luminescence was detected by a microplate reader (BioTek Synergy H1) immediately.

## Immunoblotting

Total cell lysates were prepared in RIPA lysis buffer (Beyotime Biotechnology) supplemented with 1 mM PMSF and 1× protease inhibitor cocktail (Yeasen), and nuclear protein was extracted by nuclear protein extraction kit (Solarbio) in accordance with the manufacturer's instructions. The samples were separated by SDS-PAGE and transferred to the PVDF membrane (Millipore). The membranes were incubated with the indicated primary antibodies at 4 °C overnight and then incubated with an HRP-conjugated anti-rabbit IgG antibody (Cell Signaling, Cat# 7074, 1:5000) or anti-mouse IgG antibody (Cell Signaling, Cat# 7076, 1:5000) at room temperature for 1 h. The signal was visualized by using the NcmECL Ultra kit (New Cell & Molecular Biotech) and then was exposed by using MiniChemi (SageCreation).

## Immunofluorescence

Hela cells were treated as described for the indicated time and then fixed with 4% paraformaldehyde for 25 min, permeabilized with 0.25% Triton X-100 for 10 min, and blocked in 1% BSA for 1 h at 37 °C. Then, cells were incubated with indicated primary antibodies overnight at 4 °C, and followed by incubating with donkey anti-rabbit IgG Alexa Fluor 488-labeled secondary antibodies (Abcam, Cat# ab150073, 1:200) for 1 h at 37 °C. The nuclei were counterstained with DAPI (Sigma-Aldrich) before the images were acquired using a Nikon A1R+ confocal microscope.

## In vitro cGAMP synthesis

2 μM recombinant hcGAS-ΔN was incubated with DMSO or XQ2/XQ2B in the presence of 2 μM ISD and 1 mM ATP, 1 mM GTP in a buffer containing 80 mM Tris, pH 7.5, 200 mM NaCl, and 10 mM MgCl₂ in a total volume of 200 μl at 37 °C for 2 h. The samples were diluted twenty-five times in 50 mM Tris, pH 9.0. 5 ml of the sample was loaded onto a Mono Q™ 5/50 GL column (GE Healthcare). The cGAMP was eluted with a gradient starting with 50 mM Tris, pH 9.0 reaching 2 M NaCl and 50 mM Tris, pH 9.0 over 35 column volumes with the flow of 1 ml/min. The chromatogram was obtained at 254 nm, and the cGAMP peak was analyzed by the software Unicorn™ 6.4 (GE Healthcare).

## Surface plasmon resonance (SPR)

All SPR binding studies were performed with a Biacore 8 K instrument (GE Healthcare Life Sciences). hcGAS-FL / hcGAS-ΔN (1 mg mL⁻¹) was immobilized in a CM5 chip by EDC/NHS method. The tested peptide was diluted with running buffer (20 mM HEPES, pH 7.5, 125 mM KCl,10 mM MgCl₂). All the samples contained the same concentration of DMSO (0.5%). The tested peptide was injected at a flow rate of 30 μL min⁻¹ for 200 s of association, followed by 200 s of disassociation at 25 °C. The final graphs were obtained by subtracting blank sensorgrams. The data were analyzed with Biacore Insight Evaluation software.

## Fluorescence polarization

Purified hcGAS-FL (300 nM) or hcGAS-FL (D191R) (300 nM) and FAM-labeled DNA (FAM-ISD) (40 nM) were incubated with or without XQ2 in buffer (20 mM HEPSE, pH 7.5, 125 mM KCl, 10 mM MgCl₂). Samples were dispensed in 30 μL final volumes per well into a 384-well flat-bottom black plate (Corning 3575). The polarization was detected by a microplate reader (BioTek Synergy H1) with a fluorescence

polarization filter cube (excitation wavelength: 485 ± 10 nm; emission wavelength: 528 ± 10 nm) immediately.

### In silico model of hcGAS and XQ2

The docked structure of hcGAS with XQ2 was generated by the MOE software[71]. The structure of hcGAS was extracted from the reported structure deposited in PDB protein data bank (PDB code: 4O69), and the complex structure of hcGAS with XQ2 was first generated by the docking module of the MOE package. The best-ranked docked structure was then relaxed and optimized by the dynamics module of the MOE package.

### Microscale thermophoresis (MST)

The binding ability of XQ2 with Cy5-ISD and XQ2 with human cGAS were tested using NanoTemper Monolith NT.115 instrument (20% LED, 40% MST power) at 25 °C. Serial dilutions of XQ2 were mixed with 10 nM of Cy5-ISD or 100 nM of fluorescent labeled hcGAS, incubated for 10 min at room temperature in HEPES buffer (50 mM HEPES, pH 7.5, 200 mM NaCl, 20 mM $MgCl_2$, 0.05% Tween 20) with or without 1 mM ATP and 0.5 mM GTP or 2 M $(NH_4)_2SO_4$ (Sigma-Aldrich). Excitation was optimized by varying the power to obtain fluorescence intensities above 200 counts. Data were analyzed by using the MO.Affinity Analysis software.

### In vitro phase separation assay and image acquisition

Purified hcGAS-FL and Cy3-labeled DNA (Cy3-ISD) with the indicated concentrations were mixed with DMSO (0.1%) or indicated doses of XQ2 in 96-well plates (JingAn Biological) pre-coated with 20 mg/ml BSA (Sigma). Mixtures were incubated in buffer (20 mM Tris-HCl, pH 7.5, 150 mM NaCl and 1 mg/ml BSA) with standing at 37 °C in a total reaction volume of 50 μL. Phase-separated droplets were imaged at the indicated time by using Nikon A1R+ confocal microscope with a 60× oil objective, Nikon A1 camera, and X-Cite 120LED laser. Images were analyzed using FIJI software. The number of droplets was measured as the percentage of the imaging area occupied by Cy3-labeled components, and was plotted using GraphPad Prism 8.

### Fluorescence recovery after photobleaching (FRAP)

FRAP analysis was performed on a Nikon A1R+ confocal microscope. Photobleaching of the Cy3 signal in spots of ~2-μm diameter in ~5-μm selected droplets (liquid phase condensation of hcGAS-FL and Cy3-ISD) was performed with 40% laser power for 1 s using 561-nm lasers. Time-lapse images were recorded immediately at an interval of 4 s for ~240 s after bleaching. Fluorescence intensities of regions of interest (ROIs) were quantified using Nikon NIS-Elements AR (Advanced Research) software based on the method described in the previous study[51], and were plotted using GraphPad Prism 8.

### Live-cell imaging

cGAS-knockout Hela cells reconstituted with EGFP-tagged full-length human cGAS (EGFP-hcGAS-FL) were obtained by transfecting pEGFP-C2-hcGAS-FL recombinant plasmid, and were seeded into 20-mm glass-bottom cell culture dish (NEST). Cell culture media was replaced with fresh medium for 1 h, then cells were stimulated by transfection of ISD by lipofectamine 2000 (Invitrogen) after DMSO or XQ2 (10 μM) treatment for the indicated time. The cells were rinsed three times with PBS, and live-cell imaging was performed by using Nikon A1R+ confocal microscope with a 40× oil objective, Nikon A1 camera, and X-Cite 120LED laser. Images were analyzed by FIJI software.

### Enzyme-linked immunosorbent assay

THP1 cells were seeded into 96-well plates, and cell culture supernatants were collected and analyzed by ELISA kits for human IFNβ levels (R&D Systems, Cat# DY814-05) and human CXCL10 levels (R&D Systems, Cat# DY266-05) according to the manufacturer's instructions. Mice serum was collected and analyzed by ELISA kits for mouse IFNβ levels (R&D Systems, Cat# DY8234-05) and mouse CXCL10 levels (R&D Systems, Cat# DY466-05) according to the manufacturer's protocols.

### Cell viability assays

Cell viability was measured by cell counting kit-8 (TargetMol) according to the manufacturer's instructions.

### Crystal violet staining

L929 cells were seeded into 12-well plates at a density of $5 \times 10^5$ cells per well, and living cells were stained with crystal violet staining solution (Beyotime Biotechnology) according to the manufacturer's instructions.

### Mice and isolation of BMDMs

C57BL/6 6–8 weeks old wild type (WT) mice were purchased from Gempharmatech. C57BL/6 $Trex1^{+/-}$ mice were purchased from Cyagen. $Trex1^{-/-}$ mice were generated by further mating the male and female $Trex1^{+/-}$ mice, and 8-week-old male $Trex1^{-/-}$ mice were used in the experiment. Mice were housed in groups of up to 5 mice/cage at 18 °C – 24 °C ambient temperatures with 40–60% humidity. Mice were maintained on a 12 h light/ dark cycle 6 am to 6 pm. Food and water were available ad libitum. Femurs and tibias were removed from sex-matched mice of 6–8 weeks old, and the ends were clipped. Marrow was pushed from bones by using a syringe, and cell clumps were broken apart into cell particles by pipetting in PBS, and passed through a 70 μm cell strainer. Cells were resuspended in red blood cell lysis buffer (Solarbio) for 5 min after being centrifuged with $200 \times g$ for 5 min, and were resuspended in cell culture media (DMEM) with an additional M-CSF (50 ng/ml) after centrifugation, and then were seeded into a 12-well plate. Half of the cell culture media was replaced with a fresh medium on the third day, and performed a complete media change on the fifth day. BMDMs were used for subsequent experiments after the seventh day. All the animal experiments in these studies were approved and overseen by the ethics committee of the Laboratory Animal Center of Peking University Shenzhen Graduate School in accordance with the Policy on the Care, Welfare, and Treatment of Laboratory Animals. The assigned approval or accreditation number is AP0020005.

### Pharmacokinetics

The pharmacokinetics of XQ2B was studied by detecting XQ2B in mice blood from C57BL/6. Mice ($n = 6$) were injected with XQ2B (10 mg/kg) intravenously. Whole blood samples were collected in EDTA-K2 surface-coated tubes at 0.083, 0.25, 0.5, 1, 2, 5, 7, 24 h post administration. The results were analyzed by LC-MS.

### Immunohistochemistry

Tissues were resected and fixed successively in 4% paraformaldehyde, and embedded in paraffin according to the standard procedure, and then sectioned into 10-μm slices and placed on positively charged microscope slides. Sections of the brain were subjected to immunohistochemical (IHC) staining with an anti-HSV-1 antibody (Abcam, 1:100) and HRP-conjugated anti-rabbit IgG antibody (Cell Signaling, Cat# 7074, 1:200) according to standard procedures. Sections of the heart, stomach, tongue, kidney and muscle were stained with hematoxylin and eosin. Tissue sections were visualized using an Olympus IX73 microscope (20× Objective).

### Detection of autoantibodies

Anti-nuclear antibodies were performed with 1:160 diluted sera by using a FLUORO HEPANA TEST kit (MBL BEIJING BIOTECH) according to the manufacturer's protocols.

## Flow cytometry

Cell death was measured by Annexin V-FITC/PI Apoptosis Kit (Elabscience) according to the manufacturer's instructions. Data were acquired on an Attune NxT (ThermoFisher) flow cytometer and analyzed in FlowJo.

## RNA interference

siRNA of *TREX1* (si*TREX1*) was transfected into the indicated cells by Lipofectamine RNAiMAX (Invitrogen) according to the manufacturer's instructions. Human si*TREX1*, mouse si*Trex1* and *negative control* siRNAs were purchased from Sangon Biotech. The sequences of siRNAs oligonucleotides can be found in Supplementary Table S2.

## Statistical analysis

The results were presented as mean ± standard deviation (SD). The data were analyzed with One-way ANOVA followed by Dunnett's post hoc test. Asterisk indicates that the values are significantly different (*$p < 0.05$; **$p < 0.01$; ***$p < 0.001$, ****$p < 0.0001$). Data were analyzed using GraphPad Prism software (San Diego, CA, USA).

## Reporting summary

Further information on research design is available in the Nature Portfolio Reporting Summary linked to this article.

## Data availability

Reference genomes and primers used in this study are provided in the Supplementary Information. All other relevant data that support the findings of this study are available within the article and Supplementary Information files. Source data are provided with this paper.

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

## Acknowledgements

This work was supported by funds from the NSFC (22077009 and 21521003 to J.Q.; 81973174 to X.B.), the Shenzhen Science and Technology Innovation Program (JCYJ20200109120408264 to J.Q.), the Shenzhen Fundamental Research Program (GXWD20201231165807007-20200811141635001 to J.Q.; GXWD20201231165807007-20200811151825001 to Q.L.). Guangdong Basic and Applied Basic Research Foundation (2022A1515012221 to X.B.)

## Author contributions

X.W. and J.Q. designed all experiments. X.W., Y.W., and A.C. performed the experiments. Y.W., D.C., and X.B. contributes to preparation of the macrocyclic peptide library. Q.H.L. contributes to establishment of biosensor-based high-throughput assay. Q.K.L. assisted X.W. in cellular experiments. W.Z, and J.X. contribute to structural analysis and modeling. All authors discussed the results and commented on the manuscript. J.Q. supervised all aspects of the project, and wrote the manuscript with editing and writing assistance from all authors.

## Competing interests

The authors declare no competing interests.
