## [Peer review file · Nature Communications]

Development of cyclopeptide inhibitors of cGAS targeting protein-DNA interaction and phase separationREVIEWER COMMENTS

Reviewer #1 (Remarks to the Author):

The authors identify a novel cGAS inhibitor and demonstrate that it works via inhibition of DNA-induced liquid phase condensation of cGAS. In addition, the data provided suggests that the inhibitor is specific, and also has activity in vivo. Some suggestions for further improvement.

Figure 1. In addition to evoking IFN responses, cGAS-STING signaling also activates e.g. NF- κ B, autophagy, and apoptosis. The authors should test the effect of the inhibitor on these cGAS-induced activities.

Figure 2. Since the ultimate goal is to treat diseases, the effect of XQ2 on pathway inhibition in primary human cells should be evaluated.

Figure 5. The virus replication-survival data have been made on THP1 cells. There are monocyte/macrophage-like cells, which are not a natural target for HSV1. Therefore, these data have limited physiological relevance. Rather, the authors should repeat this experimental set-up in an epithelial-like or neuronal-like cell line. Of course primary cells would be even better.

Figure 6. Does cGAS inhibitor treatment impact on disease development and survival after HSV1 infection?

Reviewer #2 (Remarks to the Author):

Previous work on the biological roles of cGAS-mediated signaling in response to cytosolic DNA has led to considerable interest in developing cGAS inhibitors. Efforts by many research groups have led to the identification of several putative cGAS inhibitors, most of which target its active site. In this paper, Wang et. al report the discovery of cyclic peptides that inhibit cGAS by interfering with DNA binding. They presented data showing that their compound XQ2b modestly inhibits cGAS activity both in vitro and in vivo. Consistent with this, they show that the compound XQ2 is capable of inhibiting phase separation of cGAS, which requires direct interaction with DNA and dramatically enhances cGAS' enzymatic activity.

A major weakness of this paper is that the cyclic peptides they identified are very weak inhibitors of cGAS, with K_d of ~ 35 micromolar. The IC_{50} values of these peptides in cultured cells are lower (which needs explanation as elaborated below), but are still quite high (>10 micromolar). As such, these are just very poor inhibitors. Even though they showed some effects of an inhibitor (XQ2b) in the Trex1 knockout mouse model, the effects were quite modest (e.g, Fig 7c may not have statistical significance due to low sample size). Another issue is that there is no data for the mode of binding between cGAS and XQ2 – the docking model shown in Figure 3c is very speculative, likely wrong and should not be presented. A real structure of cGAS bound to an inhibitor is needed for the binding mode to be convincing.

Other major issues:

1) This work is based on candidates that were identified from a relatively small, focused library of cyclic peptides derived from Gramicidin S, but there is no discussion of how this library was designed. Given that the library had a relatively high hit rate (4/90), any library design considerations should be included. Relatedly, what features, if any, did the 4 hits have in common? This could help inform future work in the field.

2) As noted by the authors, the in vivo (fig. 1b-c) inhibition of the 4 hits from the screen had no correlation with their inhibition in vitro (fig. 1a). The authors speculate that "The discrepancy between the inhibitory effect of the hits in protein and cellular assays may be attributed to their different cellular permeability." But that doesn't explain why XQ2's in vivo inhibition is stronger than its in vitro activity. Similarly, and perhaps even more surprising is that although XQ2 is reported to have a K_d of 35 μ M (in the absence of DNA) treating cells with only 10 μ M of XQ2 is

reported to abrogate the majority of cGAS signaling in many places throughout the paper, including fig. 2a and supplementary fig. s2. How is this possible? At this concentration/Kd most of the cGAS isn't expected to be bound by XQ2, especially in the presence of DNA which, according to the authors' model, competes for the same binding site.

3) What are the concentrations of cGAS and FAM-ISD in the FP assay in Fig. 3 b,d and Supplementary Fig. S3? This should be noted in the methods.

4) Appropriate errors associated with the measured Kd values of XQ2 and cGAS should be reported in the form of 95% CI or similar. This is important for readers to evaluate the authors claim that truncated cGAS binds XQ2 "with similar affinity" to full-length cGAS.

5) In fig. 2 e-f how was the quantitation of nuclear IRF3 performed in fig. 2 e-f? Ideally, this should be done algorithmically, especially given the striking difference in the IRF3 fluorescence signal in DMSO vs. XQ2 treated cells infected with HSV1.

6) As mentioned by the authors, previously described cGAS-DNA interaction inhibitors were found to intercalate into DNA. It seems important to rule out the possibility that these inhibitors are binding to DNA or to the DNA-cGAS complex instead of binding directly to cGAS.

7) Is there evidence that XQ2 and DNA bind to cGAS in a competitive manner?

Minor Issue

In supplementary fig. 1f, several compounds increased cGAMP production. Were any of those characterized?

Reviewer #3 (Remarks to the Author):

The authors describe in this manuscript the development of a cGAS inhibitor that targets the DNA/cGAS interface. The initial hit XQ2 was identified from a small cyclic peptide library, and the proposed mode of action is consistent with the disruption of DNA/cGAS binding in a fluorescent polarization assay. XQ2B, a more soluble analog, suppressed interferon and cytokine induction by HSV infection and Trex1 knockout/knockdown in vitro and in vivo. Overall, the presented data support the main claims. Given the lack of good inhibitors to curb aberrant cGAS signaling, I support publication of this manuscript after addressing the following issues.

90 cyclic peptides with limited diversity were used in the cGASi screen. The hit rate is very high (4/90). The authors indicated that these molecules are related to gramicidin S. What is the rationale for the design of such a highly focused library?

The binding mode of XQ2 was assessed by docking and supported by the lack of inhibition of the cGAS D191R mutant. However, the docking was performed on PDB 6CT9--a cGAS/DNA complex. DNA binding induces significant conformational changes in the activation loop where XQ2 is proposed to bind. The docking (or better, molecular modeling) should be reassessed with the inactive structure(s) of apo-cGAS.

Does the new docking/molecular modeling results support the D191R mutation result? Can this binding mode explain the structure-activity relationship of the 90 cyclic peptides? What is the impact on the reciprocal modification of R1' of XQ2 (e.g., changing to r or A)? Also would the R1'D analog of XQ2 restore the activity toward cGAS D191R?

A better visualization (with a 2D schematic representation?) of Fig 3c is needed for the readers to comprehend the protein/ligand interactions. Also the current Fig 3c is confusing as it looks like XQ2 binds to the DNA/cGAS complex.

Is there PK/PD data on XQ2B to support the qod dosing?

Why stopped the Trex1^{-/-} mouse study so early? There is no statistical significance in the endpoint results.

Line 59: J Org Chem 2020, pp1579 describes a cGAS inhibitor that potentially targets a nearby site at the cGAS/cGAS interface.

Line 153: This sentence is confusing: Fluorescence recovery after photobleaching (FRAP) assay showed that the fluorescence of Cy3-ISD was efficiently recovered within 30 min after mixing Cy3-ISD and cGAS (Fig. 4b, c).

Response to peer-reviewers

We would like to thank the referees for the careful reading of our manuscript, and also thank the referees for the good suggestions for improving the manuscript. We have revised the manuscript based on the comments by the referees. Generally, we have conducted several additional experiments.

To address the concern of the discrepancy between the in vitro and cellular inhibitory activity of the inhibitors, we determined the binding affinity of XQ2 with cGAS by microscale thermophoresis (MST) in the presence or absence of ATP/GTP (Fig. S4d). ATP/GTP significantly enhanced the binding affinity of XQ2 with cGAS ($K_d = 4.3 \pm 1.9 \mu\text{M}$ vs $K_d = 49.5 \pm 37.2 \mu\text{M}$), suggesting that the high levels of intracellular ATP/GTP facilitate the binding of XQ2 with cGAS. Consistently, ammonium sulfate ($(\text{NH}_4)_2\text{SO}_4$), a mimics of phosphate groups that stabilizes the activation loop of cGAS (Zhang, X., *et al. Cell Rep.* 2014, 6, 421), also markedly enhanced the binding affinity of XQ2 with cGAS (Fig. S4e). In contrast, high levels of intracellular ATP/GTP would compete with the inhibitors of cGAS targeting the active site.

To address the potential role of the inhibitors on cGAS-induced autophagy, we have performed western blotting analysis on the level of LC3-II in L929 cells induced by ISD in the presence or absence of XQ2. The level of LC3-II was increased by ISD transfection, which is significantly suppressed by XQ2 (Fig. S3).

To address the dosing of XQ2B, we have carried out pharmacokinetics studies on XQ2B in mice, the details were shown in Fig. S10.

To address the concern of the relatively short duration and small sample size of the *Trex1*^{-/-} mouse study, we repeated the in vivo study in *Trex1*^{-/-} mice in a longer period and larger samples to demonstrate a significant improvement of survival of *Trex1*^{-/-} mice treated by XQ2B (Fig. 7b, c).

All related figures have been revised according to the editorial policy.

Specifically, we would like to provide our point-by-point responses as following:

Reviewer #1

1) Figure 1. In addition to evoking IFN responses, cGAS-STING signaling also activates e.g. NF- κ B, autophagy, and apoptosis. The authors should test the effect of the inhibitor on these cGAS-induced activities.

Response: We appreciate the suggestion from the reviewer concerning the potential effect of the inhibitor on other cGAS-induced activities besides evoking IFN responses. We have demonstrated the inhibitory effect of the inhibitors on cGAS-induced NF- κ B pathway as reflected by that XQ2 and XQ2B inhibited the mRNA level of IL-6 induced by ISD or HSV-1 in a dose-dependent manner (Fig. S2a, b and S7f). Furthermore, we have examined the effect of XQ2 and XQ2B on cGAS-induced autophagy. Both XQ2 and XQ2B significantly blocked ISD-induced increases in LC3-II levels in L929 cells (Fig. S3 and S7g), highlighting the inhibitory effect of XQ2 and XQ2B on cGAS-induced autophagy. On the other hand, we did not further test the effect of the inhibitors on cGAS-induced apoptosis because the induction of apoptosis is not a general response to the activation of STING, but instead it is a cell type-specific phenomenon that is evident in primary T cells rather than other cell types (Gulen, M.F., *et al., Nat. Commun.* 2017, 8, 427).

2) Figure 2. Since the ultimate goal is to treat diseases, the effect of XQ2 on pathway inhibition in primary human cells should be evaluated.

Response: We agree with the comments of the reviewer that the effect of XQ2 on pathway inhibition in primary human cells should be evaluated. We have evaluated the inhibitory effect of XQ2 against cGAS in human and murine cell lines including human monocytic THP-1 cells, murine fibroblast L929 cells and macrophage RAW264.7 cells, we further confirmed the inhibitory effect of XQ2 against cGAS in primary bone marrow-derived macrophages (BMDMs) from C57BL/6 mice (Fig. 2a, Fig. S2c, S2d). Though currently we cannot perform the test in primary human cells due to ethical and policy issues, we noticed that primary human macrophages from human blood have similar or even higher sensitivity to cGAS inhibitors compared to THP-1 cell line in the previous study (Lama, L., *et al. Nat. Commun.* 2019, 10, 2261). This work aims to provide a proof of concept of the efficacy of novel cGAS inhibitor targeting protein-DNA interaction and phase separation, we will explore the effect of XQ2 and its analogues on cGAS inhibition in primary human cells in the future work.

3) Figure 5. The virus replication-survival data have been made on THP1 cells. There are monocyte/macrophage-like cells, which are not a natural target for HSV1. Therefore, these data have limited physiological relevance. Rather, the authors should repeat this experimental set-up in an epithelial-like or neuronal-like cell line. Off course primary cells would be even better.

Response: We agree with the comments of the reviewer that THP-1 cells are not a natural target for HSV-1, the effect of the inhibitors should be repeated in epithelial-like or neuronal-like cell line. We tested the effect of XQ2 on human cervical cancer epithelial Hela cells. HSV-1 infection did produce much higher cytopathogenic effect (CPE) in Hela cells compared to that in THP1 cells with the same virus titer, XQ2 further enhanced the CPE (Fig. R1), but the severe cell death caused by virus infection markedly interfered with the proper evaluation of immune response in Hela cells upon HSV-1 infection given that epithelial cells are generally weak responders of cGAS-STING pathway compared to immune cells like THP-1 cells (Skouboe, M.K., *et al. PLoS Pathog.* 2018, 14(4): e1006976.). On the other hand, THP-1 cells have been considered as a semi-permissive model for HSV infection though they are relatively less permissive to viral replication than other cell types such as epithelial cells (Siracusano, G., *et al. Sci Rep.* 2016, 6:31302). In addition, strong type I interferon response in THP-1 cells induced by DNA virus infection makes it a suitable cell type for extensive studies of the activation and inhibition of cGAS-STING pathway (Li, X. D., *et al. Science* 2013, 341: 1390; Lama, L., *et al. Nat Commun.* 2019, 10(1):2261).

Fig. R1. XQ2 promotes cytopathogenic effect induced by HSV-1 infection. HeLa cells and THP1 cells were pretreated for 3 h with DMSO or XQ2 (10 μ M), followed by the indicated titer of HSV-1 infection. The cell viability was measured by CCK-8 assay.

4) Figure 6. Does cGAS inhibitor treatment impact on disease development and survival after HSV1 infection?

Response: We have demonstrated that the cGAS inhibitors significantly suppressed the induction of interferon and cytokines induced by HSV-1 infection, and facilitated the infection of HSV-1 in vitro and in vivo (Fig. 5 and 6), which is consistent with the observations of other types of cGAS inhibitors that inhibit virus-induced type I interferon response and facilitate the infection of DNA viruses (Liu, Z.S., *et al. Nat Immunol.* 2019, 20(1):18; Dai, J., *et al. Cell* 2019, 176(6):1447). According to the previous studies, cGAS inhibitors would promote HSV-1-induced death and reduce survival of mice after HSV-1 infection, but we didn't explore the effect of XQ2B on the survival of mice upon HSV-1 infection according to the ethics of animal experiments that the use of animals should be minimized to ensure experimental needs.

Reviewer #2

1) A major weakness of this paper is that the cyclic peptides they identified are very weak inhibitors of cGAS, with K_d of \sim 35 micromolar. The IC_{50} values of these peptides in cultured cells are lower (which needs explanation as elaborated below), but are still quite high (>10 micromolar). As such, these are just very poor inhibitors. Even though they showed some effects of an inhibitor (XQ2b) in the *Trex1* knockout mouse model, the effects were quite modest (e.g, Fig 7c may not have statistical significance due to low sample size). Another issue is that there is no data for the mode of binding between cGAS and XQ2 – the docking model shown in Figure 3c is very speculative, likely wrong and should not be presented. A real structure of cGAS bound to an inhibitor is needed for the binding mode to be convincing.

Response: We agree with the comments of the reviewer that the binding affinity of the cyclic peptide XQ2 with cGAS in vitro is relatively weak with K_d value of \sim 35 μ M, but the cellular efficacy of XQ2 ($< 10 \mu$ M) is comparable to that of the reported cGAS inhibitors such as RU.521 (Fig. 2a, 2c in the revised manuscript), G150 (Lama, L., *et al. Nat. Commun.*, 2019, 10, 2261), compound 6 (An, J., *et al. Arthritis Rheumatol.* 2018, 70, 1807), and CU-76 (Padilla-Salinas, R., *et al. J Org Chem.* 2020, 85, 1579). The cyclic peptide XQ2 and its analogue XQ2B therefore represent a unique type of cGAS inhibitors that have higher cellular efficacy than the in vitro efficacy (Detailed explanation for this discrepancy seen in the following corresponding response for Q3 of Reviewer #2). Moreover, XQ2B, the optimized analogue with better PK and safety profiles, showed significant therapeutic effect in the *Trex1*^{-/-} mouse model (Fig. 7 in the revised manuscript).

We agree with the reviewer's comment that a cocrystal structure of cGAS bound to the inhibitor is necessary to demonstrate the detailed binding mode of the inhibitor. We have been trying hard to get the cocrystal structure of cGAS bound to XQ2 or XQ2B. Unfortunately, we only got the crystal structure of apo cGAS rather than the complex probably due to the exclusion of the inhibitor by the crystal packing (Fig. R2). On the other hand, a combination of modeling and biochemical analysis has been extensively used to understand the potential binding modes of the cGAS inhibitors when no

cocrystal structure of cGAS bound to the inhibitor is available (Padilla-Salinas, R., *et al. J Org Chem.* 2020, 85, 1579; An, J., *et al. Arthritis Rheumatol.* 2018, 70, 1807; Zhao, W., *et al. J Chem Inf Model.* 2020, 60, 3265).

Fig. R2. Crystal packing prevents XQ2 bind to hcGAS. (a) Modeled structure of hcGAS-XQ2 complex. (b) Crystal packing of hcGAS in our solved crystal structure of apo cGAS. (c) The potential steric clash between XQ2 and the crystal packing site is highlighted in the superposed structures.

2) This work is based on candidates that were identified from a relatively small, focused library of cyclic peptides derived from Gramicidin S, but there is no discussion of how this library was designed. Given that the library had a relatively high hit rate (4/90), any library design considerations should be included. Relatedly, what features, if any, did the 4 hits have in common? This could help inform future work in the field.

Response: The cyclic peptide library derived from Gramicidin S was initially developed to regulate protein-protein interactions by targeting the relatively flat and solvent-exposed protein interface (Sun, H.X., *et al. J. Med. Chem.* 2020, 63, 11286; Chen, D.Y., *et al. Chem Commun.* 2017, 53, 13340). Given the similarity between the protein-protein interface and protein-DNA interface, the cyclic peptide library was screened against cGAS-DNA interaction in this manuscript. The relatively high hit rate of the library indicated that the amphiphilic feature of XQ2 and its analogues is complementary to the DNA-binding surface of cGAS. The docking structure revealed that three of the four hits including XQ2, XQ8, and XQ19 share similar hydrophobic and electrostatic interactions with cGAS (Fig. 3c and S5b). The backbone carbonyl groups of the cyclic peptides formed four hydrogen bonds with the sidechains of Asn210, Tyr214, His217 and Lys384 of cGAS. Notably, the positively charged side chain of one arginine residue of XQ2 formed a strong salt-bridge with the negatively charged side chain Asp191 of cGAS. Replacement of the arginine residue with neutral alanine or negatively charged aspartate in the cyclic peptides markedly reduced the inhibitory activity against cGAS (Fig. S5b and Supplementary Table S1). Consistently, D191R mutation also abrogates the binding of cGAS with XQ2 (Fig. 3d and Fig. S5d).

3) As noted by the authors, the *in vivo* (fig. 1b-c) inhibition of the 4 hits from the screen had no correlation with their inhibition *in vitro* (fig. 1a). The authors speculate that “The discrepancy between the inhibitory effect of the hits in protein and cellular assays may be attributed to their different cellular permeability.” But that doesn’t explain why XQ2’s *in vivo* inhibition is stronger than its *in vitro* activity. Similarly, and perhaps even more surprising is that although XQ2 is reported to have a K_d of 35 μM (in the absence of DNA) treating cells with only 10 μM of XQ2 is reported to abrogate the majority of cGAS signaling in many places throughout the paper, including

fig. 2a and supplementary fig. s2. How is this possible? At this concentration/ K_d most of the cGAS isn't expected to be bound by XQ2, especially in the presence of DNA which, according to the authors' model, competes for the same binding site.

Response: We agree with the comments of the reviewer concerning the discrepancy between the in vitro and cellular effects of XQ2 against cGAS. To address the concern of the discrepancy, we determined the binding affinity of XQ2 with cGAS by microscale thermophoresis (MST) in the presence or absence of ATP/GTP (Fig. S4d). ATP/GTP significantly enhanced the binding affinity of XQ2 with cGAS ($K_d = 4.3 \pm 1.9 \mu\text{M}$ vs $K_d = 49.5 \pm 37.2 \mu\text{M}$), suggesting that the high levels of intracellular ATP/GTP facilitate the binding of XQ2 with cGAS. Consistently, ammonium sulfate ($(\text{NH}_4)_2\text{SO}_4$), a mimics of phosphate groups that stabilizes the activation loop of cGAS (Zhang, X., *et al. Cell Rep.* 2014, 6, 421), also markedly enhanced the binding affinity of XQ2 with cGAS (Fig. S4e). Moreover, XQ2 blocked dsDNA-induced liquid phase condensation of cGAS through disrupting the interaction between dsDNA and cGAS (Fig. 4), which facilitated TREX1-mediated degradation of cytosolic dsDNA and further suppressed the enzymatic activity of cGAS (Zhou, W., *et al. Mol. Cell* 2021, 81, 739). This novel mode of action may account for the stronger cellular inhibitory effect of XQ2 against cGAS activation compared to its binding affinity with cGAS and inhibitory activity against the binding of cGAS with dsDNA in in vitro biochemical assays.

4) What are the concentrations of cGAS and FAM-ISD in the FP assay in Fig. 3 b,d and Supplementary Fig. S3? This should be noted in the methods?

Response: The concentrations of cGAS and FAM-ISD is 300nM and 40nM, respectively, in the FP assay in Fig 3b, d and Supplementary Fig. S5. We have revised the methods according to the suggestions of the reviewer.

5) Appropriate errors associated with the measured K_d values of XQ2 and cGAS should be reported in the form of 95% CI or similar. This is important for readers to evaluate the authors claim that truncated cGAS binds XQ2 "with similar affinity" to full-length cGAS.

Response: We have revised all the measured K_d values with the 95% Confidence Intervals (95% CI) in the revised manuscript.

6) In fig. 2 e-f how was the quantitation of nuclear IRF3 performed in fig. 2 e-f? Ideally, this should be done algorithmically, especially given the striking difference in the IRF3 fluorescence signal in DMSO vs. XQ2 treated cells infected with HSV1.

Response: We used the Nikon NIS- Elements AR 5.11.03 (Advanced Research) analysis software for the data. The relative ratio of IRF3 nuclear translocation was obtained by calculating the co-location coefficients of IRF3 (green) and DAPI (blue) in Fig. 2e-f.

7) As mentioned by the authors, previously described cGAS-DNA interaction inhibitors were found to intercalate into DNA. It seems important to rule out the possibility that these inhibitors are binding to DNA or to the DNA-cGAS complex instead of binding directly to cGAS.

Response: We appreciate the suggestion from the reviewer to rule out the possibility that these inhibitors are binding to DNA or to the DNA-cGAS complex instead of binding directly to cGAS. Both SPR and MST methods have clearly demonstrated the direct binding of XQ2 with cGAS (Fig. 3a, Fig. S4b, S4d and S4e). We further examined the interaction between XQ2 and the 45-bp dsDNA ISD by microscale thermophoresis (MST). The result indicated no binding of XQ2 with ISD, suggesting no direct binding of XQ2 with DNA (Fig. S4c). In addition, the fluorescence polarization (FP) assay showed that XQ2 reduced the binding of DNA with cGAS in a dose-dependent manner, suggesting that XQ2 interrupts the formation of DNA-cGAS complex (Fig. 3b).

8) Is there evidence that XQ2 and DNA bind to cGAS in a competitive manner.

Response: The fluorescence polarization (FP) assay showed that XQ2 reduced the binding of DNA with cGAS in a dose-dependent manner, suggesting that XQ2 competes with DNA for binding to cGAS (Fig. 3b). Furthermore, the mutation D191R abrogates the binding of cGAS with XQ2, which indicated that the binding site of XQ2 locates nearby the DNA binding site on cGAS. These data suggest that XQ2 and DNA bind to cGAS in a competitive manner.

9) Minor Issue

In supplementary fig. 1f, several compounds increased cGAMP production. Were any of those characterized?.

Response: we agree with the comments of the reviewer that several compounds that increased cGAMP production warrant further investigation. This work aims to discover cGAS inhibitors rather than activators, so we did not characterize those compounds that increase cGAMP production in this manuscript, we will explore these potential agonists in the future work.

Reviewer #3

1) 90 cyclic peptides with limited diversity were used in the cGASi screen. The hit rate is very high (4/90). The authors indicated that these molecules are related to gramicidin S. What is the rationale for the design of such a highly focused library?

Response: Given the distinct features of macrocyclic peptides to target relatively shallow protein surfaces often involved in clinically important protein-protein and protein-DNA interactions, the cyclic peptide library derived from gramicidin S was initially developed to regulate protein-protein interactions by targeting the relatively flat and solvent-exposed protein interface in our previous works (Sun, H.X., *et al. J. Med. Chem.* 2020, 63, 11286; Chen, D.Y., *et al. Chem Commun.* 2017, 53, 13340). Given the similarity between the protein-protein interface and protein-DNA interface, the cyclic peptide library was screened against cGAS-DNA interaction in this manuscript. The relatively high hit rate of the library indicated that the amphiphilic feature of XQ2 and its analogues is complementary to the DNA-binding surface of cGAS as revealed by the docking structure of XQ2 bound to cGAS (Fig. 3c).

2) The binding mode of XQ2 was assessed by docking and supported by the lack of inhibition of the cGAS D191R mutant. However, the docking was performed on PDB 6CT9--a cGAS/DNA complex. DNA binding induces significant conformational changes in the activation loop where XQ2 is proposed to bind. The docking (or better, molecular modeling) should be reassessed with the inactive structure(s) of apo-cGAS.

Response: We agree with the comments from the reviewer concerning docking structure of XQ2 bound to cGAS extracted from the cGAS/DNA complex (PDB ID: 6CT9). According to the new data in the revised manuscript (Fig. S4d, e), we found that ATP/GTP or $(\text{NH}_4)_2\text{SO}_4$ (a mimic of phosphate groups) significantly enhanced the binding affinity of XQ2 with cGAS, suggesting that XQ2 may bind to the induced structure of cGAS bound to ATP/GTP intracellularly, rather than bind to the inactive structure of apo-cGAS given the high levels of intracellular ATP/GTP. Because no crystal structure of cGAS bound with ATP/GTP in the absence of DNA is available, we used the crystal structure of cGAS in complex with sulfate ions as the modeled receptor for molecular modeling (PDB ID: 4O69) (Zhang, X., *et al. Cell Rep.* 2014, 6, 421). The predicted binding mode of XQ2 was shown in Fig. 3c and Fig. S5b, which is similar to that in the previous manuscript.

3) Does the new docking/molecular modeling results support the D191R mutation result? Can this binding mode explain the structure-activity relationship of the 90 cyclic peptides? What is the impact on the reciprocal modification of R1' of XQ2 (e.g., changing to r or A)? Also would the R1'D analog of XQ2 restore the activity toward cGAS D191R?

Response: The new molecular modeling results still support the D191R mutation result as shown in Fig. 3c and Fig. S5b. The backbone carbonyl groups of the cyclic peptides formed four hydrogen bonds with the sidechains of Asn210, Tyr214, His217 and Lys384 of cGAS. Notably, the positively charged sidechain of one arginine residue of XQ2 formed a strong salt-bridge with the negatively charged sidechain Asp191 of cGAS (Fig. 3c and Fig S5b). Consistently, D191R mutation abrogates the binding of cGAS with XQ2 (Fig. 3d and Fig. S5d). Moreover, replacement of the arginine residue with neutral alanine or negatively charged aspartate in the cyclic peptides markedly reduced the inhibitory activity against cGAS (Fig. S5b and Supplementary Table S1). We noted that we cannot achieve a consensus predicted binding mode to explain the SAR of all the 90 cyclic peptides given the complexity of conformations of the cyclic peptides and the flexibility of the protein structure, a cocrystal structure of cGAS bound to the inhibitor may be necessary to demonstrate the exact binding mode of the inhibitor on cGAS. In addition, we did not observe that the R1'D analog of XQ2 such as XQ-60 restore the activity toward cGAS D191R, which could be explained by the negatively charged side chain of aspartate forms an intra-molecular salt bridge with another arginine side chain of the R1'D analog of XQ2, rather than forms the inter-molecular salt bridge with D191R of cGAS (Fig. R3)

Fig. R3. Intramolecular salt bridge between R1'D and Arg2' in XQ-60.

4) A better visualization (with a 2D schematic representation?) of Fig 3c is needed for the readers to comprehend the protein/ligand interactions. Also the current Fig 3c is confusing as it looks like XQ2 binds to the DNA/cGAS complex?

Response: To avoid the potentially misleading representation, we have revised Fig. 3c by removing the superposed DNA, and added a 2D schematic representation in supplementary Fig. S5a.

5) Is there PK/PD data on XQ2B to support the qod dosing?

Response: To address the dosing of XQ2B, we have carried out pharmacokinetics studies on XQ2B in mice, the details were shown in Fig. S10. The PK data suggests that the dosing of XQ2B (10mg/kg, q.o.d) may be suboptimal, we will explore the optimal dosing in the future work.

6) Why stopped the *Trex1*^{-/-} mouse study so early? There is no statistical significance in the endpoint results?

Response: We appreciate the comments from the reviewer concerning the short duration of the *Trex1*^{-/-} mouse study. We repeated the in vivo study in *Trex1*^{-/-} mice in a longer period and larger samples to demonstrate a significant improvement of survival of *Trex1*^{-/-} mice treated by XQ2B (Fig. 7b, c).

7) Line 59: J Org Chem 2020, pp1579 describes a cGAS inhibitor that potentially targets a nearby site at the cGAS/cGAS interface.

Response: We appreciate the suggestion from the reviewer about covering the novel type of cGAS inhibitor targeting the dimeric interface of cGAS. We have cited this elegant work in the revised manuscript.

8) Line 153: This sentence is confusing: Fluorescence recovery after photobleaching (FRAP) assay showed that the fluorescence of Cy3-ISD was efficiently recovered within 30 min after mixing Cy3-ISD and cGAS (Fig. 4b, c).

Response: We appreciate the comments from the reviewer concerning the confusing description about FRAP assay. We have revised the sentence as following: Fluorescence recovery after photobleaching (FRAP) assay showed that the

fluorescence of cGAS-Cy3-ISD condensates was efficiently recovered when bleaching was performed within 30 min after the initiation of phase separation.

REVIEWER COMMENTS

Reviewer #1 (Remarks to the Author):

The authors have improved the work in the revision. However, I have remaining concerns.

1. The authors are not right that cGAS-STING-induced apoptosis is only observed in T cells (although I agree that it is reported in the paper that they refer to). If they performed the requested experiment in monocyte-derived macrophages (or even THP1 cells) they would find these cells also die in response to STING activation. This is consistent with findings in PMID: 24139400. Therefore the requested experiment should be performed.

2. I still believe that the identified cGAS inhibitor should be tested in primary human cells. At this stage, the work is to a too large extent driven by data from cell lines.

Reviewer #2 (Remarks to the Author):

In this revision, the authors attempted to address some of my major concerns about the poor potency of the cyclic peptides and a lack of understanding of the binding mode between the cyclic peptides and cGAS. Unfortunately, they have not been able to address these significant concerns. On the potency of XQ2, which binds to cGAS with a K_d of ~ 36 micromolar in vitro, the authors presented data showing that the XQ2 binds to cGAS with a K_d of ~ 4.3 micromolar in the presence of ATP and GTP. The proposed this as an explanation for the inhibition of cGAS by XQ2 at 10 micromolar in cell culture experiments. However, in these experiments, the extent of cGAS activation by XQ2 was quite modest, in most cases less than 50% reduction of interferon-stimulated genes (ISGs). Overall, these are weak inhibitors. This is especially concerning without a structure of cGAS bound to an inhibitor. As stated in the last round of review, the docking of XQ2 to the cGAS structure is highly speculative and may not be correct. Without such a validated structure, I am not convinced by the binding mode the authors proposed. One could argue that not every published inhibitor has a co-crystal structure. While this is true, the concern about XQ2 is that it's such a weak binder of cGAS and it's a cyclic peptide that does not appear to bind to the active site of the enzyme. Having a structure is one way to be certain that this is a bona fide cGAS inhibitor and that it has a defined binding mode.

The experiment of treating *Trex1*^{-/-} mice with XQ2B is also problematic. This is how the authors described the experiment: "XQ2B (10 mg/kg) was intravenously injected into mice every other day. Six animals from each group were sacrificed on day seven to assess tissue pathology, and the rest mice were used to analyze survival for up to 11 days (Fig. 7b). As shown in Fig. 7c, 3 of 10 untreated *Trex1*^{-/-} mice died, while none of the 10 mice treated with XQ2B died during treatment ($p = 0.035$)". So they killed 6 out of a group of 10 mice on day 7, and then used the remaining 4 mice to evaluate survival. This is not the right way to do survival study. They should have a large enough group to monitor mouse survival. They also did not describe the age and sex of the mice when the treatment started. Were the *Trex1*^{-/-} mice sick when they started the treatment? Did they really reverse the disease pathology with just 3 treatments of the peptides every other day (they sacrificed the mice on day 7)? This is a very surprising result considering that the peptide is a poor inhibitor of cGAS.

Reviewer #3 (Remarks to the Author):

The authors have reasonably addressed issues raised by the reviewers. I support the publication of this revised manuscript in *Nat. Commun.*

Response to peer-reviewers

We would like to thank the referees for the careful reading of our revised manuscript, and thank the referees for the good suggestions for improving the manuscript. We have revised the manuscript based on the comments by the referees. Generally, we have conducted several additional experiments.

To address the potential role of the inhibitors on cGAS-induced cell death, we have applied a cell counting kit 8 (CCK-8) and flow cytometry to detect cell viability and cell death induced by ISD in the presence or absence of XQ2 and XQ2B. Both XQ2 and XQ2B significantly suppressed the cell death induced by cGAS activation (Fig. S3b, S8b and c).

To address the effect of the inhibitors on pathway inhibition in primary human cells, we have determined the inhibitory effect of XQ2 and XQ2B against cGAS in primary human peripheral blood mononuclear cells (PBMCs) (Fig. 2c, S7f).

Specifically, we would like to provide our point-by-point responses as following:

Reviewer #1

1) The authors are not right that cGAS-STING-induced apoptosis is only observed in T cells (although I agree that it is reported in the paper that they refer to). If they performed the requested experiment in monocyte-derived macrophages (or even THP1 cells) they would find these cells also die in response to STING activation. This is consistent with findings in PMID: 24139400. Therefore, the requested experiment should be performed.

Response: We agree with the comments of the reviewer that the activation of cGAS-STING signaling pathway induces various cell deaths including apoptosis, necroptosis, and pyroptosis (Murthy, A.M.V., *et al.*, *Cell Death Differ.*, **2020**, 27, 2989). To address the potential role of the inhibitors on cGAS-induced cell death, we have applied a cell counting kit 8 (CCK-8) and flow cytometry to detect cell viability and cell death induced by ISD in the presence or absence of XQ2 and XQ2B (Fig. R1). Both XQ2 and XQ2B significantly suppressed the cell death induced by cGAS activation (Fig. S3b, S8b and c).

Fig. R1. XQ2 and XQ2B inhibit cGAS induced cell death. (a) Jurkat cells were exposed to the indicated doses of XQ2, and then stimulated with ISD for 36h, the cell

viability was measured by CCK8 assay. **(b)** Jurkat cells were pretreated with DMSO or XQ2B (10 μ M), and then stimulated with ISD for 36h, the cell viability was measured by CCK8 assay. **(c)** Jurkat cells were treated with DMSO or XQ2B (5 μ M), followed by stimulation with ISD (2ug/ml) for 4 h. Flow cytometric analysis of apoptosis assessed by Annexin V and PI staining. Numbers in quadrants present percentages. Error bar are SD. ** $p < 0.01$, *** $p < 0.001$ (one way ANOVA, $n = 3$).

2) I still believe that the identified cGAS inhibitor should be tested in primary human cells. At this stage, the work is to a too large extent driven by data from cell lines.

Response: We agree with the comments of the reviewer that the effect of the identified cGAS inhibitors on pathway inhibition in primary human cells should be evaluated. We have thus determined the inhibitory effect of XQ2 and XQ2B against cGAS in primary human peripheral blood mononuclear cells (PBMCs) (Fig. R2). Both XQ2 and XQ2B significantly inhibit the activation of cGAS induced by ISD in primary human PBMCs (Fig. 2c, S7f).

Fig. R2. XQ2 and XQ2B inhibit the activity of cGAS in human PBMCs. **(a)** Human PBMCs were pretreated for 3h with DMSO or XQ2 (10 μ M), and then stimulated with ISD. Induction of *IFNB1* and *IL6* mRNA was measured by qPCR. **(b)** Human PBMCs were pretreated for 4h with DMSO or XQ2B (10 μ M), and then stimulated with ISD. Induction of *IFNB1* and *IL6* mRNA was measured by qPCR. Error bar are SD. ** $p < 0.01$, **** $p < 0.0001$ (one way ANOVA, $n = 3$).

Reviewer #2

1) In this revision, the authors attempted to address some of my major concerns about the poor potency of the cyclic peptides and a lack of understanding of the binding mode between the cyclic peptides and cGAS. Unfortunately, they have not been able to address these significant concerns. On the potency of XQ2, which binds to cGAS with a K_d of ~36 micromolar in vitro, the authors presented data showing that the XQ2 binds to cGAS with a K_d of ~4.3 micromolar in the presence of ATP and GTP. The proposed this as an explanation for the inhibition of cGAS by XQ2 at 10 micromolar in cell culture experiments. However, in these experiments, the extent of cGAS activation by XQ2 was quite modest, in most cases less than 50% reduction of interferon-stimulated genes (ISGs). Overall, these are weak inhibitors. This is especially concerning without a structure of cGAS bound to an inhibitor. As stated in the last round of review, the docking of XQ2 to the cGAS structure is highly speculative and may not be correct.

Without such a validated structure, I am not convinced by the binding mode the authors proposed. One could argue that not every published inhibitor has a co-crystal structure. While this is true, the concern about XQ2 is that it's such a weak binder of cGAS and it's a cyclic peptide that does not appear to bind to the active site of the

enzyme. Having a structure is one way to be certain that this is a bona fide cGAS inhibitor and that it has a defined binding mode.

Response: We agree with the comments of the reviewer that the identified cGAS inhibitors are not very highly potent inhibitors. On the other hand, we demonstrated that the cellular efficacy of XQ2 and XQ2B (< 10 μ M) is comparable to that of the currently reported cGAS inhibitors such as RU.521 (Vincent, J., *et al. Nat. Commun.*, **2017**, 8, 750), G150 (Lama, L., *et al. Nat. Commun.*, **2019**, 10, 2261), compound **6** (An, J., *et al. Arthritis Rheumatol.* **2018**, 70, 1807), and CU-76 (Padilla-Salinas, R., *et al. J Org Chem.* **2020**, 85, 1579), the detailed comparison is shown in Fig. R3. In addition, we further demonstrated that XQ2B, the optimized analogue with better PK and safety profiles, showed significant therapeutic effect in the *Trex1*^{-/-} mouse model. Overall, this work aims to provide a proof of concept of the efficacy of novel cGAS inhibitors targeting protein-DNA interaction and phase separation, we will further optimize the potency and PK profiles of the identified cGAS inhibitors in the future work.

We agree with the reviewer's comment that a cocrystal structure of cGAS bound to the inhibitor is necessary to demonstrate the detailed binding mode of the inhibitor. We will keep trying to get the cocrystal structure of cGAS bound to XQ2 or XQ2B in the future work. On the other hand, it should be noted that a combination of modeling and biochemical analysis has been extensively used to understand the potential binding modes of the cGAS inhibitors when no cocrystal structure of cGAS bound to the inhibitor is available (Padilla-Salinas, R., *et al. J Org Chem.* 2020, 85, 1579; An, J., *et al. Arthritis Rheumatol.* 2018, 70, 1807; Zhao, W., *et al. J Chem Inf Model.* 2020, 60, 3265).

Fig. R3. Comparison of the efficacy of cGAS inhibitors in vitro biochemical assays and cellular assays.

2) The experiment of treating *Trex1*^{-/-} mice with XQ2B is also problematic. This is how the authors described the experiment: “XQ2B (10 mg/kg) was intravenously injected into mice every other day. Six animals from each group were sacrificed on day seven to assess tissue pathology, and the rest mice were used to analyze survival for up to 11 days (Fig. 7b). As shown in Fig. 7c, 3 of 10 untreated *Trex1*^{-/-} mice died, while none of the 10 mice treated with XQ2B died during treatment (p = 0.035)”. So they killed 6 out of a group of 10 mice on day 7, and then used the remaining 4 mice to evaluate survival. This is not the right way to do survival study. They should have a large enough group to monitor mouse survival. They also did not describe the age and sex of the mice when the treatment started. Were the *Trex1*^{-/-} mice sick when they started the treatment? Did they really reverse the disease pathology with just 3 treatments of the peptides every other day (they sacrificed the mice on day 7)? This is a very surprising result considering that the peptide is a poor inhibitor of cGAS.

Response: We appreciate the comments from the reviewer concerning the experiment of *Trex1*^{-/-} mice treated with XQ2B. We have clarified the potential confusion about the total number of mice in the experiment in the revised manuscript as following: “XQ2B (10 mg/kg) was intravenously injected into mice every other day. Six animals from each group were sacrificed on day seven to assess tissue pathology, and the rest (**10 mice per group**) were used to analyze survival for up to 11 days (Fig. 7b). As shown in Fig. 7c, 3 of 10 untreated *Trex1*^{-/-} mice died, while none of the 10 mice treated with XQ2B died during treatment (p = 0.035).” Moreover, 8-week-old male *Trex1*^{-/-} mice were used in the experiment, which has been revised in the animal section in methods. As shown in Fig. 7d-f, 9-week-old *Trex1*^{-/-} mice developed profound inflammation in heart, stomach, tongue, kidney, and skeletal muscle compared to the wildtype mice, while XQ2B treatment significantly attenuated the inflammation. On the other hand, pathological score in *Trex1*^{-/-} mice treated with XQ2B is still markedly high compared to the wildtype mice, suggesting that XQ2B treatment ameliorates rather than reverses the systemic inflammation in *Trex1*^{-/-} mice.

REVIEWERS' COMMENTS

Reviewer #1 (Remarks to the Author):

This reviewer is convinced, and finds that the conclusions are fully supported by the data.

Reviewer #2 (Remarks to the Author):

This is the third time I reviewed this paper. Unfortunately, I don't think this paper represents a significant advance over multiple published papers that have reported development of cGAS inhibitors. Overall, all of these inhibitors, including the XQ2 compounds reported here, are quite poor especially in terms of potency. As the authors pointed out in the rebuttal letter, the compounds reported in the Lama et al paper (PMID:31113940) had better potency than the XQ2 compounds in vitro, although the cellular activities were all quite low. Considering that the authors do not have good data to support the mode of binding of their compounds to cGAS (i.e., lack of structure), it's hard to justify publication of this paper at this stage. If the authors could present more advanced compounds (e.g., better potency) and/or complex structure to illustrate the binding mode, the story would be stronger.

Response to peer-reviewers

We would like to thank all the referees for the careful reading of our manuscript, and thank the referees for their sharp comments and good suggestions for improving the manuscript. The insightful comments will significantly advance our future research in this area.

Reviewer #2 (Remarks to the Author):

This is the third time I reviewed this paper. Unfortunately, I don't think this paper represents a significant advance over multiple published papers that have reported development of cGAS inhibitors. Overall, all of these inhibitors, including the XQ2 compounds reported here, are quite poor especially in terms of potency. As the authors pointed out in the rebuttal letter, the compounds reported in the Lama et al paper (PMID:31113940) had better potency than the XQ2 compounds in vitro, although the cellular activities were all quite low. Considering that the authors do not have good data to support the mode of binding of their compounds to cGAS (i.e, lack of structure), it's hard to justify publication of this paper at this stage. If the authors could present more advanced compounds (e.g, better potency) and/or complex structure to illustrate the binding mode, the story would be stronger.

Response: We highly appreciate the reviewer's comments and concerns. Currently no specific cGAS inhibitor is approved or even in clinical trial, cGAS inhibitors with diverse modes of action are therefore very appealing in this research area. Our work reports the specific cGAS inhibitor targeting protein-DNA interaction and phase separation, and may serve as a novel scaffold for the development of therapies in the treatment of cGAS-dependent inflammatory diseases. While the desire for high potency is understandable, this work aims to present a new type of cGAS inhibitors rather than deliver an optimized drug at this stage. XQ2B displays cellular inhibitory activities against cGAS with single-digit micromolar potencies, and shows significant therapeutic effect and favorable safety profile in the *Trex1*^{-/-} mouse model, which render it a good starting point for further optimization and development of a new class of cGAS inhibitors to treat inflammatory diseases. Moreover, we agree with the reviewer's comment that the proposed binding mode of the inhibitors to cGAS is speculative, but the mode of action of the inhibitors is supported by in vitro SPR and MST binding assays, fluorescence polarization competition assay, and the site-directed mutagenesis data as shown in the manuscript. We believe that this novel and significant work will be greatly interested to the wide readers of Nature Communications.